



# A multirate mass transfer model to represent the interaction of multicomponent biogeochemical processes between surface water and hyporheic zones (SWAT-MRMT-R 1.0)

Yilin Fang[1,*], Xingyuan Chen[1], Jesus Gomez velez[2], Xuesong Zhang[3], Zhuoran Duan[1], Glenn E. Hammond[4], Amy E. Goldman[1], Vanessa A. Garayburu-Caruso[1], and Emily B. Graham[1]

[1]Pacific Northwest National Laboratory, Richland, Washington, USA

[2]Vanderbilt University, Nashville, Tennessee, USA

[3]Joint Global Change Research Institute, Pacific Northwest National Laboratory, College Park, Maryland, USA

[4]Sandia National Laboratories, Albuquerque, New Mexico, USA

**Correspondence:** Yilin Fang (yilin.fang@pnnl.gov)





**Abstract.**

Surface water quality along river corridors can be modulated by hyporheic zones (HZs) that are ubiquitous and biogeochemically active. Watershed management practices often ignore the potentially important role of HZs as a natural reactor. To investigate the effect of hydrological exchange and biogeochemical processes on the fate of nutrients in surface water and
HZs, a novel model, SWAT-MRMT-R, was developed coupling the Soil and Water Assessment Tool (SWAT) watershed model and the reaction module from a flow and reactive transport code (PFLOTRAN). SWAT-MRMT-R simulates concurrent non-linear multicomponent biogeochemical reactions in both the channel water and its surrounding HZs, connecting the channel water and HZs through hyporheic exchanges using multirate mass transfer (MRMT) representation. Within the model, HZs are conceptualized as transient storage zones with distinguished exchange rates and residence times. The biogeochemical pro-
cesses within HZs are different from those in the channel water. Hyporheic exchanges are modeled as multiple first order mass transfers between the channel water and HZs. As a numerical example, SWAT-MRMT-R is applied to the Hanford Reach of the Columbia River, a large river in the United States, focusing on nitrate dynamics in the channel water. Major nitrate contaminants entering the Hanford Reach include those from the legacy waste, irrigation return flows (irrigation water that is not consumed by crops and runs off as point sources to the stream), and groundwater seepage resulted from irrigated agriculture.
A two-step reactions for denitrification and an aerobic respiration reaction are assumed to represent the biogeochemical transformations taking place within the HZs. The spatially variable hyporheic exchange rates and residence times in this example are estimated with the basin-scale Networks with Exchange and Subsurface Storage (NEXSS) model. Our simulation results show that 1) as the commonly used transient storage model for stream–HZ exchange of solutes uses a single residence time to parameterize the exchange coefficient, it may overestimate the nitrate attenuation role of HZs ignoring the contribution from
HZs with low residence times; and 2) source locations of nitrate have different impact on surface water quality due to the spatially variable hyporheic exchanges.

# 1 Introduction

Broadly defined, the hyporheic zone (HZ) is the area of the stream bed and stream bank in which stream water mixes with shallow groundwater (Runkel et al., 2003), or through which subsurface pathways begin and end at the stream (Cardenas,
2015). The HZ has been recognized as a critical component of stream ecosystem (Boano et al., 2014; Boulton et al., 1998; Ward, 2016; Wondzell, 2011; Liao and Cirpka, 2011). It is a location of interacting physical, chemical, and biological systems (Ward, 2016). Biogeochemical gradients in HZs can significantly impact cycling of carbon and nitrogen (Claret and Boulton, 2009; Briggs et al., 2013). Most studies of HZ focused on low-order streams (e.g. Briggs et al., 2009; Harvey et al., 2013; Hoagland et al., 2017; Mulholland et al., 1997; Triska et al., 1989; Valett et al., 1996; Ward et al., 2016) because they are
more logistically tractable. On the other hand, the role of HZ for large or higher order streams is not well understood due to inadequate or difficulty in direct measurements (Tank et al., 2008; Ye et al., 2017; Zarnetske et al., 2012), and the challenge to scale up the process understanding from plot scale measurements to reach scale. For large rivers, numerical experiments may provide some insights on the role of HZs.





Hyporheic exchange and biogeochemical processes within HZs are largely absent in reach-scale models (Helton et al., 2010).

We need a modeling framework that integrates the physical and biogeochemical processes in the surface water and HZs in order to bridge the gap in reach-scale models and answer questions of when and where HZs become important. Hydrologic exchange between surface water and groundwater was commonly modeled by the transient storage model (Bencala and Walters, 1983; Briggs et al., 2009; Boano et al., 2014; Runkel et al., 2003; Haggerty and Gorelick, 1995; Zaramella et al., 2003). The transient storage model is a lumped model representing first-order exchange between the main channel and well-mixed storage zones or

dead zones, usually parameterized with conservative tracer test. The transport of solute between the storage zone and channel is simply determined by the solute concentration difference between the channel and storage zone and an exchange coefficient (Bencala and Walters, 1983). It has been modified to include various zones with one exchange coefficient within each zone (Briggs et al., 2010; Neilson et al., 2010), and to simulate reactive solute within the storage zones by coupling with a chemical equilibrium model (Runkel et al., 1996); however, the transient storage model is not capable of representing the long solute

tail observed in the stream (Cardenas, 2015; Painter, 2018). Coupled with a transient storage zone solute transport model,Ye et al. (2012) used a dynamic hydrologic network model to simulate dissolved nutrient retention processes during transient flow events at the channel network scale. They assumed a constant exchange rate and simple uptake of nutrient in the storage zone, but the interplay between mass exchange and biogeochemical processes as well as hydrological and geological control on mass exchange rates were ignored.

In this study, to model the exchange between the channel and HZs, we use the multirate mass transfer (MRMT) formulation incorporating a spectrum of transition times that is commonly used to simulate the late-time solute and contaminant tails observed in porous media (e.g., Haggerty and Gorelick, 1995; Wang et al., 2005; Liu et al., 2008; Fernandez-Garcia and Sanchez-Vila, 2015). The MRMT formulation is a discrete equivalent to convolution-based representations to model mass exchange, which can be solved in Lagrangian domain (Silva et al., 2009). Painter (2018) recently generalized the convolution-

based model to include multicomponent reactive transport with general nonlinear reactions and tested the model using a single reach, assuming a steady state hyporheic flow field. It has not yet been integrated in watershed modeling. Here we generalized the MRMT model to include mass transfer between the stream channel and multiple storage zones according to a spectrum of rates within each storage zone. We developed an integrated model, referred to as SWAT-MRMT-R, based on two open-source codes to simulate in-stream biogeochemical processes, mass transfer between the stream and HZ, and biogeochemical

processes in HZs. These two codes are the Soil and Water Assessment Tool (SWAT) watershed model (Neitsch et al., 2011) and the subsurface flow and reactive transport code, PFLOTRAN (Lichtner et al., 2017). We applied the integrated model to the Hanford reach of the Columbia river, a large river in the southern Washington State of the United States. At the simulated reach, major impacts to stream nutrients come from the contaminated groundwater, irrigation return flows, and groundwater seepage resulted from irrigated agriculture (Evans et al., 2000).

This paper starts with an overview of the components in the SWAT-MRMT-R modeling framework, followed by description of each component (Sect. 2). The model is then applied at the Hanford reach using the spatially variable hyporheic exchange rates and residence times estimated with the basin-scale Networks with Exchange and Subsurface Storage (NEXSS) model





(Gomez-Velez and Harvey, 2014) (Sect. 3). We focus on the nitrate dynamics in the results (Sect. 4), varying modeled processes and parameters. Finally, model limitation and future work are discussed.

## 70 2 SWAT-MRMT-R Model Description

### 2.1 Modeling Framework

Our modeling framework is built within the in-stream nutrient submodel in SWAT, considering distinguished biogeochemical processing in both the water column and hyporheic zones. The SWAT model is developed by the U.S. Department of Agriculture to predict the impact of land management on water, nutrients, and sediments of large ungauged basins (Neitsch et al.,
2011) and it is widely used worldwide. A schematic description of the modeling framework is shown in Fig. 1. At the top of the figure, the processes pointed to by the blue arrow are reactions related to the water column. The processes in the orange circle at the bottom of the figure are example reactions related to the hyporheic zones (shown as black rectangular prisms with orange streamlines) in the vertical and lateral directions. The orange streamlines represent the river water entering the vertical and lateral hyporheic zones and returning back to the river within the same reach over a distribution of travel times,
i.e., each streamline carries a residence time ($\tau_s$) and exchange flux ($q_s$). In this study, the residence time and exchange flux are estimated by NEXSS (Gomez-Velez and Harvey, 2014), which will be briefly described in Sect. 3.3. We use an MRMT model to represent solute mass transfer dynamics between the channel and storage zones, capturing the effect of hyporheic exchange in the fate and transport of solutes along the river corridor. All of the reactions and exchange processes shown in Fig. 1 are simulated simultaneously with the implicit time stepping through the Newton Raphson method in batch mode (i.e., no
transport) of PFLOTRAN (Lichtner et al. (2017). In the following, we describe each of these components and their coupling in detail.

### 2.2 Mass Exchange Processes between the Channel and Storage Zones

MRMT is commonly used to simulate the late-time solute and contaminant tails in porous media (e.g., Haggerty and Gorelick, 1995; Wang et al., 2005; Liu et al., 2008; Fernandez-Garcia and Sanchez-Vila, 2015). We use it here to simulate mass transfer
between the stream channel and multiple storage zones according to a spectrum of rates within each storage zone. The mass exchange rate of a solute component due to hyporheic exchange is:

$$m_{s,j} = (C_j - C_{s,j})q_s L, \tag{1}$$

where $j$ is the solute component of interest ($O_2$, $NO_3^-$, DOC, and $NO_2^-$ etc.), $s = 1, ..., N$ is the storage zone, $m_{s,j}$ is the mass exchange rate of the $j$-th component [$MT^{-1}$], $C_j$ is the concentration in the main channel [$ML^{-3}$], $C_{s,j}$ is the concentration
of the $j$-th component in the $s$-th storage zone [$ML^{-3}$], $q_s$ is water exchange flux per reach length [$L^3 L^{-1} T^{-1}$], and L is the length of the reach.





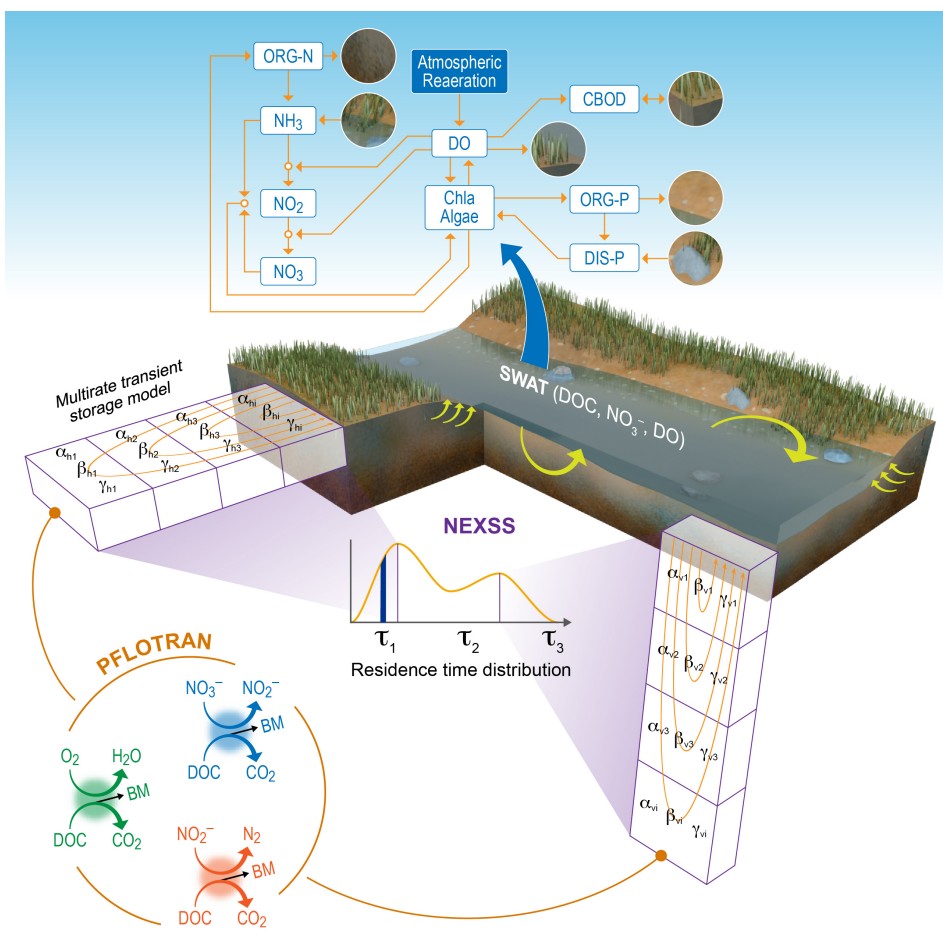

**Figure 1.** Schematics of coupling SWAT, MRMT, and microbial reactions in HZs. Top of the figure shows reactions in the stream water column. In the circle are representative reactions in the HZs. Rectangular prisms are sub-storage zones. The orange curves in the rectangular prisms are streamlines. Residence time distributions for lateral and vertical HZs are estimated by NEXSS. $\alpha_i$ is the exchange rate corresponding to residence time $\tau_i$ in the sub-storage zone $i$. $\beta_i$ and $\gamma_i$ represent reaction rates for different reactions in the sub-storage zone $i$. All chemical transformation processes shown are solved using PFLOTRAN.

### 2.3 Mass Balance Equations of the Coupled System

Considering the biogeochemical processes in the water column, storage zones, and hyporheic exchange, the final mass balance equations for dissolved components in the channel and storage zones are:


$$AL\frac{\partial C_j}{\partial t} + \sum_{s=1}^{N} m_{s,j} = AL\Big(D\frac{\partial^2 C_j}{\partial x^2} - u\frac{\partial C_j}{\partial x} + \sum_{m=1}^{M} a_{m,j} r_m\Big), \tag{2}$$





$$A_s L \frac{\partial C_{s,j}}{\partial t} = m_{s,j} + A_s L \sum_{ms=1}^{Ms} a_{ms,j} r_{ms}. \tag{3}$$

The area of the storage zones are calculated as a function of each storage zone's water exchange flux ($q_s$) and storage residence time ($\tau_s$):

$$A_s = q_s \tau_s, \tag{4}$$

where $A$ is the cross-sectional area of the channel [$L^2$], $A_s$ is the cross-sectional area of the storage zone $s$ [L$^2$], $t$ is time [T], $\tau_s$ is storage residence time [T], $r_m$ is the reaction rate of the $m$-th ($m = 1, ..., M$) reaction in the channel water [ML$^{-3}$T$^{-1}$], $a_{m,j}$ is the stoichiometry of the $j$-th component in the $m$-th reaction $i$ [-], $r_{ms}$ is the reaction rate of the $ms$-th ($ms = 1, ..., M_s$) reaction in the storage zone [ML$^{-3}$T$^{-1}$], $a_{ms,j}$ is the stoichiometry of the $j$-th component in the $ms$-th reaction $i$ [-], $D$ is longitudinal dispersion [L$^2$T$^{-1}$], and $u$ is flow velocity in the channel [LT$^{-1}$]. The other variables are defined in Eq. (1).

Equations (1)-(4) are a parsimonious version of the general form of the MRMT model (e.g., Haggerty and Gorelick, 1995; Anderson and Phanikumar, 2011) where a discrete number of storage zones, each with different geometry, linear rate, and biogeochemical reactions, are considered.

### 2.4 Water Column Biogeochemical Processes

The in-stream submodel within SWAT is based on the QUAL2E (Brown and Barnwell, 1987) model. QUAL2E simulates major
interactions of the nutrient cycle, atmospheric aeration, algae production, benthic oxygen demand, and carbonaceous oxygen uptake (Fig. 1). Details of the processes and transformation rates can be found in Brown and Barnwell (1987) and Neitsch et al. (2011). Major processes shown in Fig. 1 in the channel water column are summarized below:

1. Algae growth and respiration. Algae grow via photosynthesis and die via respiration. As algae grow and die, they form an organic part of the in-stream nutrient cycle. Nutrient limitation factor is considered for algal growth.

2. Nitrogen cycle. The transformation from organic nitrogen to ammonia, to nitrite, and finally to nitrate is stepwise.

3. Phosphorus cycle. Organic phosphorus is mineralized to soluble phosphorus available for uptake by algae.

4. Transformation of dissolved oxygen (DO). Concentrations of DO can be changed due to atmospheric reareation, photosynthesis, plant respiration, benthic demand, biochemical oxygen demand, and nitrification.

Algae, organic nitrogen, and phosphorus can also be removed from the stream by settling. All of the reactions described above
contribute to $r_m$ in Eq. (2).





## 2.5 Storage Zone Biogeochemical Processes

Although any type of biogeochemical process of interest can be simulated by the model, we focus on the following reactions for aerobic respiration and denitrifiction in each storage zone:

$$R1: CH_2O + f_1O_2 + \frac{1}{5}(1-f_1)NH_4^+ \rightarrow f_1CO_2 + \frac{1}{5}(1-f_1)C_5H_7O_2N + \frac{1}{5}(3+2f_1)H_2O + \frac{1}{5}(1-f_1)H^+$$

$$R2: CH_2O + 2f_2NO_3^- + \frac{1}{5}(1-f_2)NH_4^+ \rightarrow 2f_2NO_2^- + f_2CO_2$$
$$+ \frac{1}{5}(1-f_2)C_5H_7O_2N + \frac{1}{5}(3+2f_2)H_2O + \frac{1}{5}(1-f_2)H^+$$

$$R3: CH_2O + 2f_3NO_2^- + \frac{1}{5}(1-f_3)NH_4^+ \rightarrow \frac{2}{3}f_3N_2 + f_3CO_2$$
$$+ \frac{1}{5}(1-f_3)C_5H_7O_2N + \frac{1}{15}(9+16f_3)H_2O + \frac{1}{15}(3+17f_3H^+), \tag{5}$$

where the reactions R1, R2, and R3 are associated with the electron acceptors $O_2$, $NO_3^-$, and $NO_2^-$, respectively, and $f_1$, $f_2$, and $f_3$ are the fractions of the electron equivalents in $CH_2O$ used for energy production in each reaction. We used the approach proposed by Song et al. (2017) and Song et al. (2018) to model the reaction rates for $R1$, $R2$, and $R3$ (denoted as $r_1$, $r_2$, and $r_3$, which contribute to $r_{ms}$ in Eq. (3):

$$r_i = e_i r_i^{kin}, i = 1, 2, 3. \tag{6}$$

These reaction rates incorporate unregulated and regulated effects. In particular, the unregulated effect is represented by a Monod-type kinetics coefficient ($r_i^{kin}$) of the form

$$r_i^{kin} = k_i \frac{a_i}{K_{a_i} + a_i} \frac{d_i}{K_{d_i} + d_i} [BM], \tag{7}$$

where the $k_i$, $K_{a,i}$, and $K_{d_i}$ denote the reaction rate [mol mol $^{-1}$ $d^{-1}$], half-saturation constants associated with the electron acceptors [mol $L^{-1}$], and half-saturation constants for donors [mol $L^{-1}$], respectively, $a_i$ is the concentration of electron acceptor [mol $L^{-1}$], and $d_i$ is the concentration of electron donor (mol $L^{-1}$), [BM] is the concentration of biomass [mol $L^{-1}$].

On the other hand, the regulated effect is dictated by the enzyme activity ($e_i$) and determined by the cybernetic control law (Young and Ramkrishna, 2007) as follows (Song et al., 2018):

$$e_i = \frac{r_i^{kin}}{\sum_i^3 r_i^{kin}}. \tag{8}$$

This approach associates a community's traits with functional enzymes without directly considering the dynamics of individual microbial species or their guilds that synthesize their enzymes (i.e., the entire microbial community is treated as a single





organism) and allowed the estimation of reaction stoichiometries and rates for denitrification, as well as biomass degradation rate (Song et al., 2017). Rather than using empirical inhibition kinetics, the formulation in Eq. 8 enables the oxygen regulation of denitrification in response to environmental variation (Song et al., 2018).

## 2.6 Numerical Solution of the Nonlinear Reaction System

In SWAT, dissolved nutrients are transported with the water and those sorbed to sediments are allowed to be deposited with the sediments on the bed of the channel (Neitsch et al., 2011). Nutrient transport and reactions in SWAT are solved sequentially. We modified the explicit time-stepping algorithm in the original code for in-stream chemistry so the resulting nonlinear system of equations from the transformations taking place within the stream water and storage zones are simulated simultaneously with the implicit time stepping through the Newton Raphson method in batch mode (i.e., no transport) of the PFLOTRAN

(Lichtner et al., 2017) model. PFLOTRAN is an open source, massively-parallel reactive multiphase flow and multicomponent transport code. It has well-established documentation (https://www.pflotran.org/documentation/).

During each time step, the initial condition in the water column was calculated using the following equation (Neitsch et al., 2011):

$$C_{j,i} = \frac{m_{j,wb} + m_{j,flowin}}{V_{stored} + V_{flowin}},$$
(9)

where $C_{j,i}$ is the initial concentration of dissolved component $j$ in the water [ML$^{-3}$] within reach $i$, $m_{j,wb}$ is the amount of component $j$ in the water body at the beginning of time step [M], $m_{j,flowin}$ is the amount of component $j$ added to the water body with inflow at the end of time step [M], $V_{stored}$ is the volume of water stored in the channel at the beginning of time step [L$^3$], and $V_{flowin}$ is the volume of water entering the channel at the end of time step [L$^3$], calculated using the Muskingum routing method (Overton, 1966).

## 165 3 Numerical Experiment

### 3.1 Site Description

We tested our model in a ninth-order reach of the the Upper Columbia-Priest Rapids watershed located in southern Washington State (N46.23∼46.86, W118.14∼120.25). The selected reach is adjacent to the Hanford site downstream of the Priest Rapids Dam, representing a regulated river section with significant agricultural nutrient inputs. The Priest Rapids watershed has a

semi-arid climate with a Mediterranean precipitation pattern. Winters are cold, with a mean temperature of ca. -2.2$^o$C. Water is pumped from the river for irrigation and runoff is returned to the river through canal drains. This watershed is dominated by agricultural activities on irrigated land. The Columbia River is the primary source of surface water for irrigation. Major crops in the watershed include alfalfa, potatoes, spring barley, winter wheat, corn, and orchards. Sources of pollutants entering the Columbia River are irrigation return flows and groundwater seepage associated with irrigated agriculture. Dilution in the river

results in contaminant concentrations that are below drinking water standards.





## 3.2 SWAT Model Configuration

We set up the SWAT model by combining a series of geospatial data. We derived topography information from the U.S. Geological Survey (USGS) National Elevation Dataset (https://lta.cr.usgs.gov/NED) with a spatial resolution of 30 meters. Land covers including shrubland, forestland, grassland, developed land, barren land, and cultivated land were derived from

the US Department of Agriculture Cropland Data Layer (https://nassgeodata.gmu.edu/CropScape/) with a spatial resolution of 30 meters. Figure 2a shows the shapes of the reaches within the watershed. A reach is a section of stream between two defined points. Figure 2b compares the coarser reaches (green lines) generated for SWAT with those (black lines) defined in the National Hydrography Database (NHD Plus V2, http://nhd.usgs.gov) at a selected location. NHD Plus V2 is used by NEXSS for residence time and exchange flux estimations. Highlighted in cyan is the main channel of the Columbia River.

Climate data for the period of 2001-2015 from North America Land Data Assimilation System (https://ldas.gsfc.nasa.gov/nldas/NLDAS2forcing.php) was used to obtain precipitation, temperature, relative humidity, solar radiation, and wind speed. Discharge of water and nitrate at the USGS 12472900 station near the Priest Rapids Dam were used as inputs. In addition, we obtained nitrogen and phosphorus fertilizer application rates (USDA-ERS, 2018), tillage intensity (CTIC, 2008), and planting and harvesting (USDA, 2010) for crop management (Qiu and Malek, 2019).

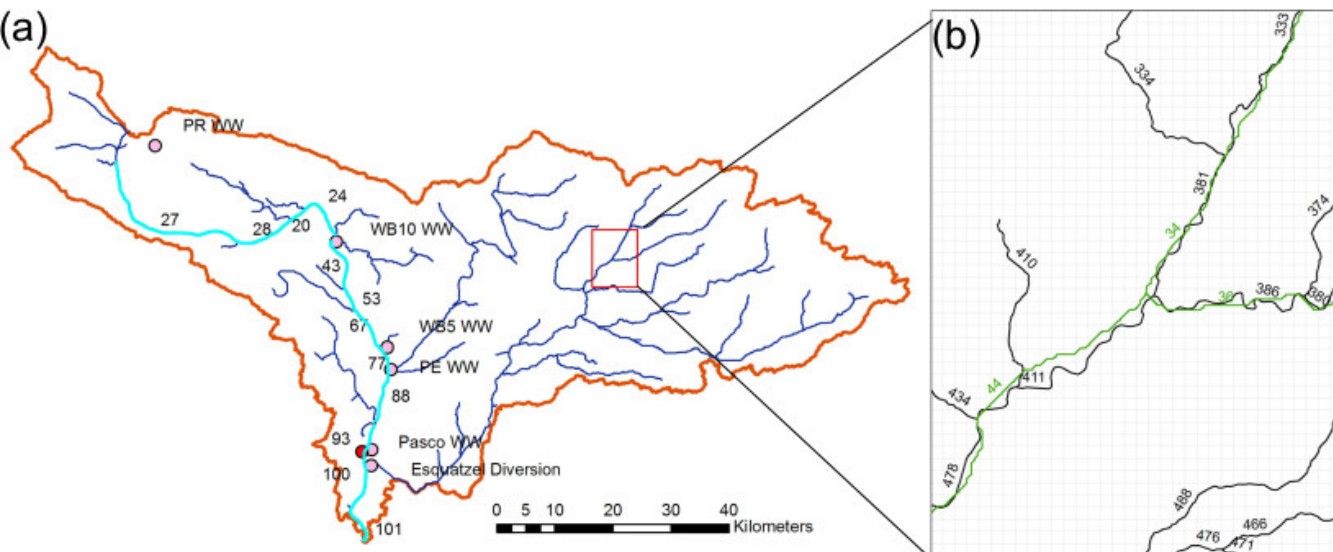

**Figure 2.** Reaches created by SWAT, highlighted in cyan is the main channel of the Columbia River (a). Numbers identify the individual reaches, the red circle is the area where field samples for dissolved organic carbon were collected, pink circles are the main wasteways carrying irrigation return flow water from farms and irrigation system operations, and the red rectangle shows the selected location for the mapping between SWAT and NHD Plus v2 that's used for NEXSS simulation. (b) shows the mapping between NEXSS (black lines) and SWAT (green lines) for the rectangle area. The numbers in (b) are reach identifications.



## 3.3 Mass Exchange Parameters

Quantifying hyporheic exchange in streams is difficult. Exchange rate coefficients are often determined by tracer injection in the stream (e.g., Harvey et al., 2013; Gooseff et al., 2011; Laenen and Bencala, 2001). Assuming steady uniform flow, the river corridor hydrologic exchange model NEXSS in Gomez-Velez and Harvey (2014) can estimate hydrologic exchange flux rates and distribution of residence times that can be used to parameterize the exchange rate coefficients. NEXSS is a single tool that consolidates multiple previously published geomorphic and hyporheic flow models for the analysis of exchange at the basin scale (Gomez-Velez et al., 2015). The river network is discretized into individual reaches of length L, where values of channel width, bankfull width, depth, bankfull depth, discharge, average channel velocity, median grain size, channel slope, sinuosity, and regional hydraulic head gradient in the x and y direction are prescribed. The model is applied in two steps: geomorphic characterization and hyporheic exchange modelling. For each channel reach, the hyporheic exchange modelling predicts average hyporheic exchange flux per unit area of streambed and residence time distributions, as well as median residence time for vertical exchange beneath submerged bedforms (ripples, dunes and alternate bars) and lateral exchange outside the wetted channel through emergent alternate bars and meanders. To estimate the net amount of vertical exchange flux, vertical exchange is conceptualized as a three-dimensional process, assuming a homogeneous and isotropic porous medium. The boundary conditions of the system are: the top boundary at the sediment-water interface has a prescribed head distribution, lateral boundaries are impervious, and the bottom boundary has a prescribed flux. The lateral exchange is conceptualized as steady and two-dimensional process, and the flow is solved by the vertically integrated groundwater flow equation with the Dupuit–Forchheimer assumption. A particle tracking scheme is used to estimate the flux-weighted residence time distribution and median residence time for the exchange zones. Detailed description of NEXSS and its assumptions can be found in Gomez-Velez and Harvey (2014). Consistent with Gomez-Velez et al. (2015), we considered two storage zones corresponding to lateral and vertical exchange. NEXSS estimates these parameters for long-term average flow conditions and mean monthly flow conditions and therefore it can only capture the seasonal dynamics of flow.

Note that additional preprocessing is needed to match the scales of NEXSS and the model proposed in this study. NEXSS simulations are performed within the National Hydrography Database, which is characterized by reach lengths that range from 23 m to 43 km within the watershed of interest. This is different from the discretization used in the stream network delineated for the SWAT simulation (Fig. 2b). To address this issue of mapping the NEXSS exchange metrics into the SWAT stream network, we implemented a procedure that matches channels and transfers parameters over a rectangular grid that covers the entire watershed. As the reaches from the two networks do not exactly overlap (see Fig. 2b), we used a grid size of 200 m to search for matching reaches. Reaches that intersected with this grid and the length of each reach within each grid cell is calculated. For example, NEXSS reaches 386 and 380 in (Fig. 2b) contribute to SWAT reach 36. To calculate the exchange flow for SWAT reach 36, exchange flux from NEXSS 386 and 380 are multiplied by the area of the channel bed and added together. The residence time for SWAT reach 36 is then calculated using residence time weighted by exchange flow of each contributing NEXSS reach. NEXSS reaches that do not overlap those of SWAT are not considered. Input file to the code includes: 1) reach identification number, vertical and lateral exchange fluxes, vertical and lateral residence times, reach width, and reach length





within a grid from NEXSS; and 2) reach identification number, reach length within a grid from SWAT. Binary trees are created

for both of the stream networks for NEXSS and SWAT to facilitate the search. When more than two reach segments of the two networks landed in the same grid cells, the matching reaches are found.

Mapping the NEXSS output to the simulated watershed, we found that the reaches along the Columbia River are characterized by vertical residence times ranging from 8 hours to 10 days, and the lateral residence times ranging from 17 days to 10 years (Fig. 3). The vertical and lateral residence times for the reaches highlighted in Fig. 2a are shown in Table 1 in the

Supplement. Compared to other reaches in the watershed, the Columbia River is characterized by relatively larger exchange flow and shorter residence times in vertical storage zones (Fig. 3). Hydraulic conductivity, head gradient, stream geomorphology, and flow paths together determine the hyporheic residence times (Kasahara and Wondzell, 2003; Cardenas et al., 2008). The vertical exchange residence times are short, probably due to their hydraulic and geomorphic conditions that favor ripple and dune formation (Gomez-Velez et al., 2015). For example, hyporheic flow around steps and riffles with coarse-textured

alluvium can result in short residence time distributions for hyporheic exchange flows (Kasahara and Wondzell, 2003). Vertical exchange flow is dominant along the Columbia River.

### 3.4  Parameters for Biogeochemical Reactions

Parameters for reactions in the stream water are default from SWAT. The selection of reaction parameters for HZs are informed by the denitrification experiments and modeling synthesis reported in Li et al. (2017) and Song et al. (2017), respectively. Stegen

et al. (2016) showed that significant $O_2$ maintained in the HZ at the location shown by the red circle in Fig. 2a, which implies that aerobic respiration can take place in the HZ. Therefore, aerobic respiration and denitrification reactions are considered in this study. The parameters for biogeochemical reactions are the same as in Song et al. (2018), which were based on parameters derived from the aforementioned experiment as well as parameters for aerobic respiration assuming it is energetically more favorable than $NO_3^-$ reduction. The uptake rates ($k_i$ in Eq. 7) of oxygen, nitrate and nitrite are 68.94, 28.26, and 23.28 mmol

mmol$^{-1}$ d$^{-1}$, respectively. Biomass degradation rate is 0.242 mmol mmol$^{-1}$ d$^{-1}$. The other parameters are not repeated here.

### 3.5  Case Scenarios

Starting from the base case, where mass transfer and reactions in the HZ were not simulated, we added mass transfer with only one exchange rate calculated from the NEXSS mean residence time for each zone. Then reactions in the HZs were added to isolate the impact of physical and biological processes on nitrate level in the river in order to answer the questions of the role

of HZs and biogeochemical processes in HZs to nitrate concentration attenuation in the surface water. These three cases are referred to as BASE, MRMT, and MRMT+BGC, respectively. Additional cases with exchange and BGC include: 1) replacing the residence time and exchange flux with those predicted by NEXSS using seasonal flow conditions; 2) replacing the single storage zone in vertical and lateral with sub-storage zones within a storage zone, assuming a distribution of residence time to evaluate whether the commonly used single transient storage model is sufficient to model nitrate attenuation in the river; 3)

pulsed increase of nitrate in the hyporheic zones to see how surface water quality will be affected when pulsed nitrate from the





**Figure 3.** NEXSS simulated vertical (a,c) and lateral (b,d) residence times (h) and exchange flow rate (m$^3$ $s^{-1}$) in the watershed, and scatter plot of exchange flow (m$^3$ $s^{-1}$) versus residence time (h) (e). Solid circles in (e) are for reaches along the Columbia River.





legacy waste intrude into the HZs that could be caused by a flow event; and 4) adding irrigation return flows along the river as point sources to study if the river can quickly dilute the contaminant concentration from those sources because of the high discharge rate.

For case MRMT+BGC, the initial conditions for DOC, nitrate, and DO in the HZs were set to 0.0637 mmol $L^{-1}$, 0.079 mmol
$L^{-1}$, and 0.287 mmol $L^{-1}$, resepectively, the average concentrations of those observed in the field. Initial BM concentration is 0.01 mmol/L. As there is not a working model for DOC in the version of SWAT (SWAT rev664) for this study, we fitted an exponential relationship of DOC and stream discharge (shown in the supporting information Fig. S1) in the Columbia River using the limited measurement of DOC in the stream and stream discharge at the time of the measurements. This relationship is used to calculate DOC in the stream.

To represent multiple exchange rates according to a residence time distribution, the discrete form of the probability density function of residence time is written as

$$p(\tau_s) = \sum_j P_j \delta(\tau_s - \tau_{s,j}), \tag{10}$$

where $\tau_{s,j}$ is the residence time of the $j$-th sub-storage zone, $P_j$ is the probability of occurrence of the residence time in the $j$-th sub-storage zone such that

$$\sum_j P_j = 1 \tag{11}$$

$\tau_{s,j}$ can be determined from the following equation using interpolation method:

$$f_j = \int_{\tau_s - \Delta\tau_s}^{\tau_s + \Delta\tau_s} p(\tau_s) d\tau_s, \tag{12}$$

in which $f_j$ is defined as the fraction of the sub-storage zone within the vertical or horizontal storage zones that has an average residence time $\tau_s$. Here we assume equal fraction for each sub-storage zone, i.e., a vertical or horizontal storage is evenly divided into $N_s$ sub-storage zones.

## 4 Results

### 4.1 MRMT with single vertical and lateral storage zones

Compared to the BASE case, pure mass exchange by MRMT has little impact on the downstream nitrate level in the river due to the small exchange flow compared to the stream discharge (Fig. 4a). Vertical HZ reached dynamic steady state in the selected reach, but the concentration in the lateral HZ is still decreasing (Fig. 4b,c), indicating continuous losing of concentrations in the HZ due to slow mass transfer.

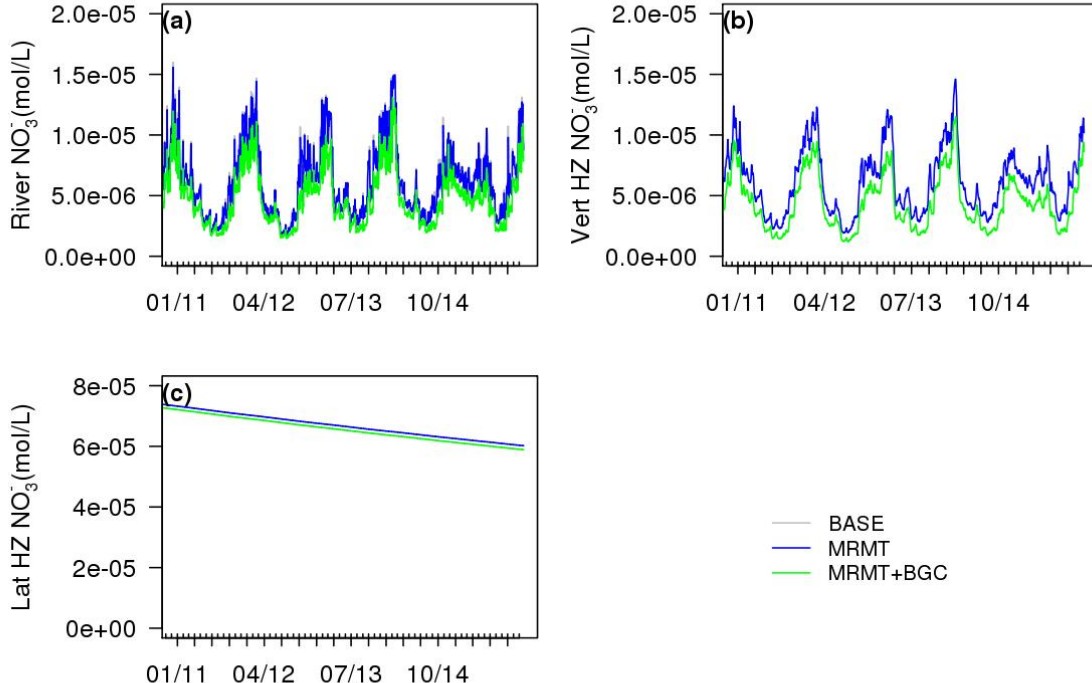

**Figure 4.** Comparison of nitrate in the river (a), vertical HZ (b), and lateral HZ (c) between cases BASE, MRMT, and MRMT+BGC at reach 93.

Figure 5 shows the concentration of nitrate in selected reaches (from upstream to downstream) along the Columbia River for case MRMT. The selected reach numbers are shown in Fig. 2a. In general, nitrate concentrations in the stream and HZs are nearly identical for reaches with similar residence times. Exchange with HZs of longer residence time slows the transport

of nitrate in the stream (retardation) as can be inferred from the delayed peak concentration in vertical HZs of reaches RCH53 and RCH101 in Figs. 5B,D. Stream nitrate concentrations are in dynamic steady state with the vertical HZ. They are also in dynamic steady state with the lateral HZ for reaches RCH27, RCH24, RCH28, and RCH20, suggesting that lateral exchange can be important too. It's not true for RCH77 and RCH88 as their exchange flows are much smaller.



**Figure 5.** Nitrate concentration in the stream and HZs along the Columbia River.

## 4.2 HZ Microbial Respiration

Including the respiration reactions in the simulation results in consumption of stream nitrate (green line in Fig. 4a) because the reaction time is faster compared to vertical HZ residence times in some reaches along the river, but on the similar order. Nitrate concentration in the vertical HZ at the selected reach (RCH93) drops due to the respiration reactions, creating a concentration difference between the channel and HZs, further decreasing the concentration in the channel compared with case MRMT. Lateral HZ is diffusion limited (slow exchange), hence not much dynamics in nitrate concentration (Fig. 4c).



### 4.3 Seasonal exchange flux and residence time

For this case, dynamic exchange parameters were generated using NEXSS driven by long-term averaged monthly stream discharges. During the simulation, exchange parameters were updated each month. Figure 6a,b,c shows the change of vertical exchange flux and residence time for RCH101 (outlet) as stream discharge changes: 50% change of stream discharge results in about 12% change of exchange flux and residence time. Using the seasonal exchange fluxes does not have a significant effect on the stream water nitrate concentration (Fig. 6d,e,f). However, it increases the nitrate concentration in the vertical HZ in the fall as the small exchange rates slow down the nitrate and carbon delivery to the HZ, reducing the reaction rates.

### 4.4 Multiple sub-storage zones

Zhang and Baeumer (2007) showed that the long tail behavior in stream tracer can be approximated by a small number of mass transfer coefficients. We started with 20 mass transfer coefficients or sub-storage zones for both the lateral and vertical HZs. Using exponential residence time distribution and 20 sub-storage zones, we had multiple rates based on the mean residence time from NEXSS. Assuming the exchange flux from the NEXSS estimation is equally distributed in each sub-storage zone, the residence time for each sub-sotrage zone is calculated using Eq. 12. Simulation with multiple exchange rates within each storage zone showed less removal of nitrate in the stream through microbial respiration in the HZs compared to the single-rate simulation (Fig. 7). Model results show no difference with 10 or 20 sub-storage zones, except for much longer simulation time for 20 sub-zones.

### 4.5 Pulsed nitrate increase in HZs

Contaminant can enter the stream during the upwelling of nitrate-contaminated groundwater from the legacy waste site. We changed the nitrate concentration to 0.3 mmol $L^{-1}$ in the vertical HZ of RCH27 and RCH101 separately at the beginning of 2011. Because the residence time in HZ of RCH27 is 30 times faster than RCH101 (Table 1 in the Supplement), the stream and HZ nitrate concentration reached steady state instantaneously (Fig. 8a,b) and the high concentration pulse propagated downstream (Fig. 8c,d) because of the longer residence time downstream. Local perturbation at the outlet (RCH101) results in a small change in stream water concentration because of the longer residence time (Fig. 8c,d). At the outlet, it takes 112 days for the reach concentration to return to a pre-perturbed condition. It has been observed in the field that high stream nitrate concentration than those shown in the bASE case can occur. The field observation may be explained by the contaminant intrusion as tested here. The non-equilibrium behavior shown in this test indicates that dynamic mass contribution from the groundwater should be considered in this model as it affects concentrations in the HZs.

### 4.6 Irrigation return flows

There are five major wasteways carrying return flow water into the Columbia River (Fig. 2a). Bureau of Reclamation, Pacific Northwest Region, has recorded nitrate loading from these wasteways. We generated an hourly nitrate loading (Fig. 9) as five



**Figure 6.** Effect of streamflow dynamics (a) on hyporheic exchange (b,c), and nitrate concentration in the stream (d), vertical HZ (e), and lateral HZ (f) using exchange rates derived from annual discharge (blue line) and seasonal discharge (green line) at the outlet




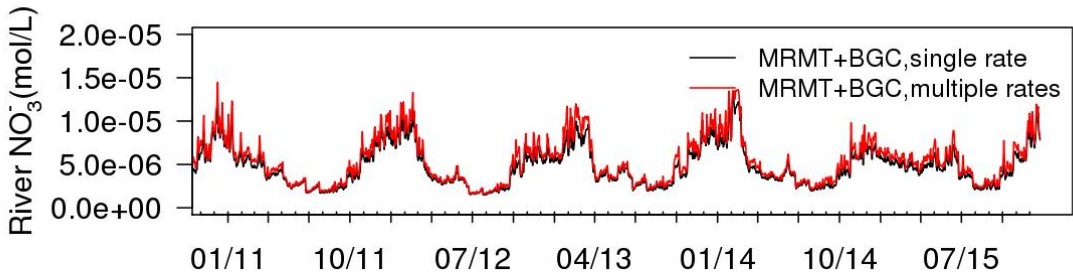

**Figure 7.** Nitrate concentration in the stream at reach 93 using single (black line) and multiple exchange rates (red line) within each HZ.

source points and reran the MRMT+BGC case. Wasteways WB5 and PE flow into RCH77 and RCH88, respectively. These two reaches have short residence times, thus nitrate coming from these wasteways will exchange in the HZs in a short time and will not be expected to have a big impact on surface water quality, as it is shown that the increase of nitrate concentration in the main channel at RCH77 almost equals to the increase in vertical HZ (Fig. 10a,b) compared to the case without return flows. Wasteways WB10, Pasco, and Esquatzel Diversion flow into RCH43, RCH93, and RCH100, respectively. These reaches have relatively longer residence times, which means nitrate coming from these wasteways will stay in the stream and deteriorate downstream water quality. Water treatment attention should be paid to these wasteways before they flow into the Columbia River. Compared to the case without return flows, Fig. 10c,d shows there is more increase (as high as 20%) of nitrate concentration in the main channel caused by return flow from the Esquatzel Diversion (the highest nitrate loading source) compared to that in the vertical HZ at RCH101 because of the long residence times in RCH100 and RCH101.

## 5 Discussion

Our study focuses on the effect of bedform-induced (vertical) and sinuosity-driven (lateral) hyporheic exchange derived under steady-state conditions on stream water quality in large rivers using the SWAT-MRMT-R model we developed. Our studied system is dominated by inlet conditions if there are no additional sources in the HZs.

### 5.1 Coupled effect of physical and biogeochemical processes

Coupling reactive transport in channel and HZs, as well as MRMT in a large river using the mean residence times and exchange fluxes calculated from NEXSS, our simulations show that HZs can attenuate the peak nitrate concentrations in the stream with mass transfer and biogeochemical reactions (Fig. 4). However, for biogeochemically inactive zones, hyporheic exchange





**Figure 8.** Nitrate concentration in the stream and vertical HZ at RCH27 (a),(b) and at the outlet (c),(d) with and without HZ perturbation.



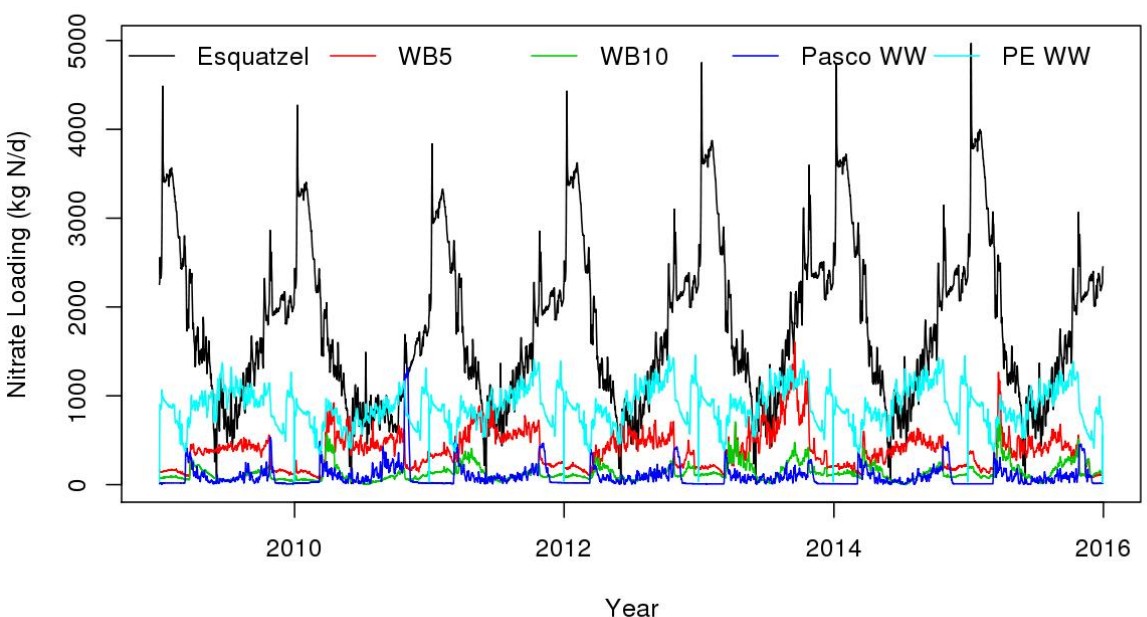

**Figure 9.** Nitrate loadings at five wasteways along the Columbia River.

(physical process) alone is not effective in attenuating nitrate in the surface water due to the relatively small exchange flow in the vertical HZs. Depending on the channel morphologic features and the river channel aquifer's physical properties of an

individual river corridor, lateral HZ can play an important role in storage and biogeochemical transformation in certain reaches as the vertical HZ (Fig. 5), which echoes the findings in Gomez-Velez et al. (2015).

## 5.2   Effect of dynamic hyporheic exchange

Our simulation using seasonally dynamic hyporheic exchange shows that, although not significant, there is some decrease in nitrate removal in the channel compared to the constant exchange case (Fig. 6d) caused by the low exchange fluxes in the

fall. The Hanford Reach of the gravel-bed Columbia River exhibits frequent hydrologic mixing and steep physicochemical gradients caused by the combined effects of large, long-term river stage fluctuations driven by snowmelt runoff and short-term fluctuations driven by upstream flow regulation at the Priest Rapids Dam (Graham et al., 2017; Stegen et al., 2016; Song et al., 2018). Gravel-bed river corridors, which are common worldwide (Nilsson et al., 2005; Sawyer and Cardenas, 2009; Hauer et al., 2016), exhibit coarse-textured sediments with high hydraulic conductivity. Hydrologic exchange due to dam operation

and high hydraulic conductivity can drive river water into and out of the river banks, creating high frequency lateral exchanges in these type of river corridors. The study in Zhou et al. (2017) indicated the exchange fluxes in the shallow water near the river







**Figure 10.** Difference of nitrate concentration with and without irrigation return flows in the stream and vertical HZ at RCH 77 (a),(b), and at the outlet (c),(d).





banks of the Columbia River were stronger than those in the center of the channel due to a large pressure gradient between the groundwater and river stage. This is not captured by the steady-state NEXSS model, which could result in an inaccurate estimation of the attenuation capability by the HZs and recovery time for the stream water quality.

## 5.3 Effect of multirate mass transfer using multiple rates

Short residence times could provide faster supply of DOC into the HZs. Including the short residence time can frequently change the chemical signature of the stream water in the HZ and could be used to explain, for example, the response of microbial community composition in Stegen et al. (2016). The importance of the biogeochemical processes in HZs on surface water quality could be affected by the fraction of this short residence time if Damköhler number for oxygen ($Da_{O2}$) is greater than 1 (Zarnetske et al., 2012). Our simulation with multiple rates shows that channel nitrate removal could decrease compared to a single rate (Fig. 7). This conclusion is caused by the equal exchange flux assumption for each sub-storage zone that decreased the fraction of short residence time compared to the single rate case in our simulation and the steady state NEXSS assumption. It has yet to be verified by field studies.

## 5.4 Implications for source water quality treatment

Spatial perturbation in the HZs and including point sources of irrigation return flow show that high-concentration pulses of nitrate may stay in the stream before being attenuated (Figs. 8 c,d and 10c,d). This has implications for the effort to treat water quality in groundwater and irrigation return flows before entering the reaches with longer residence times, particularly for streams impacted by agricultural activities.

## 5.5 Limitations and future development

There are limations in the current model such as 1) the lack of nitrification processes, 2) source uncertainties, e.g., lack of groundwater contribution to the HZs in the model; 3) uncertainty in the exchange rates and residence times calculated by NEXSS; and 4) uncertainty in the rates for biogeochemical reactions in the HZs. For example, the shape of residence time distributions can be drastically changed due to heterogeneity in stream sediment hydraulic conductivity as shown in the study of Pryshlak et al. (2015). We only considered denitrification processes in our study case, which is reasonable in this study as the Damköhler number for oxygen ($Da_{O2}$) is > 1 based on residence times calculated from NEXSS and oxygen uptake rate assumed. If $Da_{O2}$ becomes less than 1, nitrification processes which dominate short residence times (Zarnetske et al., 2012) should be included in the model to better quantify the role of HZs as a net sink of nitrate. These processes can be easily implemented in PFLOTRAN. Also, we only evaluated the denitrification process using a single set of reaction rate parameters in the river corridors. However, nutrient uptake in streams can be highly spatially variable (McClain et al., 2003). Our perturbation and return flow simulations show that source term can have a big impact on stream water quality. Effect of HZs on stream water quality can be compromised by upwelling of nitrate-contaminated groundwater through adding nitrate to





the stream or altering features of the hyporheic zone (Azizian et al., 2017). These factors need to be evaluated and combined with field characterization and model development in future studies.

## 6   Conclusion

Based on widely used open-source models SWAT and PFLOTRAN, we developed an integrated model, SWAT-MRMT-R, to account for hyporheic exchange and biogeochemical processes within HZs that are often absent in reach scale modeling. In this model, the MRMT module can incorporate a spectrum of transition times to represent solute mass transfer dynamics between the channel and storage zones. The model is flexible such that different shape of residence time distributions (e.g., exponential, power-law, or lognormal distribution) and biogeochemical reactions in stream water and HZs can be easily implemented.

Exchange parameters in our study were estimated from NEXSS. Although our demonstration of the model uses mass transfer parameters derived from steady state discharge conditions, it is applicable to dynamic conditions as well once the corresponding mass transfer parameters are available. An integrated model as developed in this study can serve as a base framework to answer questions related to the significance of HZs in surface water quality in higher order river networks where characterization of HZs can be challenging. Our simulation results show that apart from the well known lack of long tail representation of tracer,

the commonly used transient storage model may overestimate the contaminant attenuation role of HZs. Effort are needed to treat water quality in groundwater and irrigation return flows before entering the reaches with longer residence times.

*Code and data availability.*   The model code (SWAT-MRMT-R v1.0) (Fang et al., 2019a) and the data (Fang et al., 2019b) used to produce the results in this study are available on the Zenodo repository. Zenodo DOIs for the model and data are 10.5281/zenodo.3585948 and 10.5281/zenodo.3585976, respectively.

*Author contributions.*   YF developed SWAT-MRMT-R, which incorporates model code PFLOTRAN previously developed by GEH. JG generated output from NEXSS and ZD postprocessed the NEXSS output. XZ set up the SWAT model. YF did all model simulations, prepared the figures, and wrote the paper with support from XC. XC also helped design the simulations. AEG, VAG, and EBG provided the field data in the Supplement.

*Competing interests.*   The authors declare that they have no conflict of interest.

*Acknowledgements.*   This research was supported by the U.S. Department of Energy (DOE), Office of Biological and Environmental Research (BER), as part of BER's Subsurface Biogeochemistry Research (SBR) program. This contribution originates from the SBR Scientific Focus Area at Pacific Northwest National Laboratory (PNNL). This paper describes objective technical results and analysis. Any subjective





views or opinions that might be expressed in the paper do not necessarily represent the views of the U.S. Department of Energy or the United States Government. PNNL is operated for the DOE by Battelle Memorial Institute under contract DE-AC05-76RL01830. Sandia National

Laboratories is a multimission laboratory managed and operated by National Technology  Engineering Solutions of Sandia, LLC, a wholly owned subsidiary of Honeywell International Inc., for the U.S. Department of Energy's National Nuclear Security Administration under contract DE-NA0003525. A portion of this research was performed using Institutional Computing at PNNL.



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
