# Peer review of "A multirate mass transfer model to represent the interaction of multicomponent biogeochemical processes between surface water and hyporheic zones (SWAT-MRMT-R 1.0)"

_Geoscientific Model Development, 2019_

## Short Comment (SC1) · 5 Mar 2020

The discussion paper "A multirate mass transfer model to represent the interaction of multicomponent biogeochemical processes between surface water and hyporheic zones (SWAT-MRMT-R 1.0)" presents a coupling between the SWAT watershed model and the biogeochemical reaction modeling capabilities within the PFLOTRAN code. Coupling between flowing surface water and biogeochemically active hyporheic zones is a key consideration in the development of more mechanistic representations of nutrient transport and transformation at watershed scales. Multiscale approaches like

those adopted in the discussion paper (see also Painter 2018) represent promising strategies for incorporating more detailed biogeochemical understanding in catchment- and basin-scale models. In particular, the attempt to account for a range of hyporheic residence times and the effect of hyporheic zone flowpath diversity on net nutrient processing is welcome.

However, the discussion paper is unclear and appears to be internally inconsistent on how the hyporheic zone and its coupling to the flowing channel are conceptualized and represented in software. The schematic in Figure 1 shows computational cells (sub-storage zones in their terminology) that are connected in series, which would approximate a one-dimensional advection-dispersion-reaction system for each storage zone, as in Painter [2018]. That is, the sub-storage zone closest to the channel is exchanging mass with the channel, but other sub-storage zones are exchanging mass with their neighboring sub-storage zones, not directly with the channel. However, the description of mass exchange with multiple sub-storage zones in the caption of Figure 1 and in text in Sections 2.3 and 4.4 implies sub-storage zones that are each connected to the channel – i.e. sub-storage zones connected in parallel to the channel, which is the transient storage zone model generalized to multiple storage zones. The distinction is important. If connected in series, then the reaction system for a sub-storage zone will have as input the reaction products from neighboring sub-storage zones. If connected in parallel, then each would see only unreacted river water as input. Net nutrient processing for the integrated system is likely to be different for the two configurations because the parallel configuration would result in more mixing with oxygen-rich river water and thus underpredict redox zonation and the effect on redox-sensitive reactions.

Additionally, if the conceptualization is meant to be that of sub-storage zones connected in series, as in Figure 1, then there is an additional question of how reaction products are returned to the channel. Figure 1 implies that reaction products from one cell would pass back through the cells closer to the channel, which would be appropriate for hyporheic zones that are diffusion dominated. If hyporheic exchange is due to

advective pumping, then reaction products from sub-storage zones should be returned directly to the stream channel (see, e.g. Figure 1 of Liao et al. [2013] or Figure 1 of Painter [2018]). Again, the distinction is likely to be important for net biogeochemical processing.

In short, it would be useful to clarify the conceptualization of the spatial structure of each storage zone, its coupling to the river channel, and the representation of that process in the numerical model. If each sub-storage zone is connected directly to the channel, as implied by the text, then Figure 1 should be redrawn to represent that particular mesh topology. Additionally, a discussion of the two different ways to conceptualize the transient storage zones and their anticipated effect on simulated biogeochemical processing in stream/river corridors would be valuable to readers.

Scott Painter Environmental Science Division Oak Ridge National Laboratory

Liao, Z., Lemke, D., Osenbrück, K., and Cirpka, O. A. ( 2013), Modeling and inverting reactive stream tracers undergoing two-site sorption and decay in the hyporheic zone, Water Resour. Res., 49, 3406– 3422, 10.1002/wrcr.20276.

Painter, S.L. (2018) Multiscale Framework for Modeling Multicomponent Reactive Transport in Stream Corridors, Water Resources Research, 54, 10, (7216-7230).
* * *

---

## Referee Comment (RC1) · Anonymous Referee #1 · 26 Mar 2020

The discussion paper "A multirate mass transfer model to represent the interaction of multicomponent biogeochemical processes between surface water and hyporheic zones (SWAT-MRMT-R 1.0)" introduces a newly-developed model, named SWAT-MRMT-R, to simulate the hydrological and biogeochemical interaction between surface water and hyporheic zones. Hyporheic zone is usually ignored in large-scale watershed models because of its complex processes. Therefore, I think adding the simulation of hyporheic zone processes and connection hyporheic zones and surface water has a good intention and is appreciated. I have some major comments for this manuscript: 1.

[Figure]

The manuscript description about SWAT-MRMT-R, the connection between SWAT and MRMT, the connection of SWAT code and PFLOTRAN is not very clear. There is abundance of model concepts, software, and codes involving in this work (SWAT, MRMT, PFLOTRAN, NEXSS, NHD plus V2), and the connection between them was mentioned in different places in the manuscript. But I want to put in in a figure/schematization, to make the readers clear about the role of each of them in the model. I recommend modifying figure 1 or having a separate figure showing the connection between these. 2. It is not very clear to me about the connection in hydrology between SWAT and MRMT. Does SWAT provide inflow of upland catchment to hyporheic zones which is represented by MRMT, and how the transmission losses through streambed calculated by SWAT affect the hyporheic zones? I think hydrological exchange between surface water and hyporheic zones is very important for the constituent exchange, so please make it clear. 3. How many storage zones in your conceptualization of the hyporheic zones? Is the number of storage zones defined by the user in the setup? What is the maximum number of storage zones? 4. Why can NEXSS not use the reach system generated by SWAT to calculate residence time and exchange flux? 5. For matching the channel systems between NEXSS and SWAT, it seems that NEXSS reaches that do not overlap with those of SWAT are not considered. In the example in figure 2b, only reaches 386 and 380 contribute to SWAT reach 36. However, I think reach 374 also contributes to SWAT reach 36. If we change the threshold stream definition in SWAT, SWAT can generate a more detailed stream network and can overlap with all of the reaches from NEXSS. 6. The paper shows several modelling experiments. But I wonder if the authors did compare the model predictions with field measurements. I did not see calibration/validation in this paper. I do think the comparison with measurements is very necessary for a modelling paper. So please add a session on this if data is available. 7. Do you have a SWAT model application in this case study? Do you consider comparing SWAT-MRMT-R with SWAT to show the better prediction when hyporheic processes are included? 8. SWAT-MODFLOW coupling SWAT and MODFLOW models can represent hyporheic zones and its interaction with the stream. The

author should add reference about SWAT-MODFLOW in the manuscript. What is the advantage of SWAT-MRMT-R compared to SWAT-MODFLOW? I suggest adding a section discussing about the strengths, weakness/limitations of the models, what cases the users should use SWAT-MRMT-R in the manuscript.

―――――――――――――――――――

---

## Referee Comment (RC2) · Anonymous Referee #2 · 17 Apr 2020

The manuscript addresses the transport of nitrogen in river networks, and proposes an extension of the SWAT model that includes water exchange and biogeochemical reaction in the hyporheic zone. The model is applied to the watershed of the Columbia River. Proposing an extention of SWAT that includes the effect of hyporheic process is a valuable contribution to research in solute fate in catchments, as catchment-scale models usually do not encompass these processes. However, additional efforts are needed to clarify some methodological parts. The main issues are described below.

Main comments:

- ROLE OF GROUNDWATER: Vertical upwelling of groundwater through the hyporheic zone and to the stream seems not to be included (see detailed comments, line 205). The contribution of groundwater as lateral flow should be better explained (206). - USE OF RESIDENCE TIME DISTRIBUTION: the authors explore the impact of using a distribution of residence time instead of an average value. This is a valuable attempt, but the way these distributions are chosen is unclear (251, 305).

Detailed comments:

line 205 "the bottom boundary has a prescribed flux" -> how is it chosen? Is it equal to zero?

206 "flow is solved by the vertically integrated groundwater flow equation with the Dupuit–Forchheimer assumption" -> for consistence, I suggest to briefly state boundary conditions also for determining lateral flow.

230 "Compared to other reaches in the watershed, the Columbia River is characterized by relatively larger exchange flow" -> Fig. 3 shows exchange flow values up to the order of 100 m3/s (vertical flow) and 1 m3/s (lateral flow). These values are extremely high and deserve some explanation on why they are considered realistic. It is possibly that they occur only on very long reaches, but I would check this and verify the values of flux per unit area that provide values whose magnitude can be more easily assessed.

247 "only one exchange rate [...] for each zone" -> I would state more clearly that two zones are used to represent vertical and lateral exchange, as it comes out later.

251 "1) replacing the residence time and exchange flux with those predicted by NEXSS using seasonal flow conditions; 2) replacing the single storage zone in vertical and lateral with sub-storage zones within a storage zone, assuming a distribution of residence time [...]" -> a few additional words would make easier to understand how these scenario have been built. Specifically, I recommend to specify 1) which "seasonal flow conditions" have been considered and 2) how the characteristics of the "distribution of

GMDD
residence time" have been determined. There is a mathematical explanation for it, but it is unclear how the specific values of tau\_s,j have been chosen. At present, some of (but not all) these information are provide in sections 4.3 and 4.4 of the Results' section, but they would be better located here rather than among the Results. Alternatively, is should be at least anticipated that they are reported later.

288 "It's not true [the fact that lateral HZ are in dynamic steady state] for RCH77 and RCH88 as their exchange flows are much smaller" -> why only these reaches? RCH67, 53, 93, 100 and 101 all exhibit the same behavior.

Fig. 5, caption: it should specified that this is the MRMT scenario. Same for other figures.

305 "Using exponential residence time distribution and 20 sub-storage zones, we had multiple rates based on the mean residence time from NEXSS. Assuming the exchange flux from the NEXSS estimation is equally distributed in each sub-storage zone, the residence time for each sub-sotrage zone is calculated using Eq. 12." -> this explanation is unclear. How has each residence time estimated with NEXSS (denoted here as T) transformed into multiple (20) residence times? From this description I imagine that for each zone an expontial distribution with mean equal to T was defined, and then 20 values with their corresponding probability were extracted from this distribution. Anyway, this is not fully clear from the text, and in any case many details are missing (e.g., how was the maximum residence time chosen?). I strongly recommend to provide more details about this part as it is fundamental to obtain a representative distribution of residence times. As a further notice, the choice of equally distributed fluxes is simple but debatable, as it is known from the classic theory of Elliott and Brooks (1997) that exchange flowpaths with higher fluxes penetrate deeper in the streambed and are hence characterized by longer times. I am unsure if this would entail a significant difference. but if it is feasible I recommend verifying the impact of this assumption.

307 "sotrage" -> "storage"
307 "Simulation with multiple exchange rates within each storage zone showed less removal of nitrate in the stream through microbial respiration in the HZs compared to the single-rate simulation (Fig. 7)." -> as far as I can see by eye, the difference is rather small. If so, I would mention it.

Fig. 6: because all these scenarios include biogeochemical reactions, I recommend labeling them coherently, i.e., MRMT+BGC, SEASONAL MRMT+BGC.

318 "high stream nitrate concentration than those shown in the bASE case can occur" -> is this a regular feature or has it only been observed occasionally?

319 "bASE" -> "BASE"

325 "nitrate coming from these wasteways will exchange in the HZs in a short time and will not be expected to have a big impact on surface water quality" -> the link between residence time and impact on water quality is not so evident. What is clear from fig.10 is that in this reach HZ and river concentrations exhibit syncronous variations, as already expected from the previously shown results. I understand that if the residence time in the HZ is large enough then the increase in NO3 concentration due to the point source can be buffered and possibly attenuated, but the comments here do not clarify this well enough.

341 "our simulations show that HZs can attenuate the peak nitrate concentrations in the stream" -> it would be useful to report a quantative assessment (e.g., concentration reduction between xxx and yyy %) instead of just sending back the reader to fig.4.

375 "limations" -> "limitations"

GMDD

---

## Author Comment (AC1) · 11 May 2020

**SC1: Scott Painter**

The discussion paper "A multirate mass transfer model to represent the interaction of multicomponent biogeochemical processes between surface water and hyporheic zones (SWAT-MRMT-R 1.0)" presents a coupling between the SWAT watershed model and the biogeochemical reaction modeling capabilities within the PFLOTRAN code. Coupling between flowing surface water and biogeochemically active hyporheic zones is a key consideration in the development of more mechanistic representations of nutrient transport and transformation at watershed scales. Multiscale approaches like those adopted in the discussion paper (see also Painter 2018) represent promising strategies for incorporating more detailed biogeochemical understanding in catchmentand basin-scale models. In particular, the attempt to account for a range of hyporheic residence times and the effect of hyporheic zone flowpath diversity on net nutrient processing is welcome.

However, the discussion paper is unclear and appears to be internally inconsistent on how the hyporheic zone and its coupling to the flowing channel are conceptualized and represented in software. The schematic in Figure 1 shows computational cells (substorage zones in their terminology) that are connected in series, which would approximate a onedimensional advection-dispersion-reaction system for each storage zone, as in Painter [2018]. That is, the sub-storage zone closest to the channel is exchanging mass with the channel, but other sub-storage zones are exchanging mass with their neighboring substorage zones, not directly with the channel. However, the description of mass exchange with multiple sub-storage zones in the caption of Figure 1 and in text in Sections 2.3 and 4.4 implies sub-storage zones that are each connected to the channel - i.e. sub-storage zones connected in parallel to the channel, which is the transient storage zone model generalized to multiple storage zones. The distinction is important. If connected in series, then the reaction system for a sub-storage zone will have as input the reaction products from neighboring sub-storage zones. If connected in parallel, then each would see only unreacted river water as input. Net nutrient processing for the integrated system is likely to be different for the two configurations because the parallel configuration would result in more mixing with oxygen-rich river water and thus underpredict redox zonation and the effect on redoxsensitive reactions.

Additionally, if the conceptualization is meant to be that of sub-storage zones connected in series, as in Figure 1, then there is an additional question of how reaction products are returned to the channel. Figure 1 implies that reaction products from one cell would pass back through the cells closer to the channel, which would be appropriate for hyporheic zones that are diffusion dominated. If hyporheic exchange is due to advective pumping, then reaction products from sub-storage zones should be returned directly to the stream channel (see, e.g. Figure 1 of Liao et al. [2013] or Figure 1 of Painter [2018]). Again, the distinction is likely to be important for net biogeochemical processing. In short, it would be useful to clarify the conceptualization of the spatial structure of each storage zone, its coupling to the river channel, and the representation of that process in the numerical model. If each substorage zone is connected directly to the channel, as implied by the text, then Figure 1 should be redrawn to represent that particular mesh topology. Additionally, a discussion of the two different ways to conceptualize the transient storage zones and their anticipated effect on simulated biogeochemical processing in stream/river corridors would be valuable to readers. Scott Painter Environmental Science Division Oak Ridge National Laboratory Liao, Z., Lemke, D., Osenbrück, K., and Cirpka, O. A. (2013), Modeling and inverting reactive stream tracers undergoing two-site sorption and decay in the hyporheic zone, Water Resour. Res., 49, 3406–3422, 10.1002/wrcr.20276. Painter, S.L. (2018) Multiscale Framework for Modeling Multicomponent Reactive Transport in Stream Corridors, Water Resources Research, 54, 10, (7216-7230).

**Response:**

Thanks for the comment and discussion.

MRMT in this paper is an extension of the commonly used transient storage model to represent riverine solute transport assuming each storage zone is well mixed. The stream water column and the hyporheic zones can be conceptualized as separate batch reactors gaining or losing mass due to hydrologic exchange as shown in the schematics below. The sub-storage zones are in parallel and assumed to be well mixed. They don't communicate with each other, but communicate with the stream water, which reacts itself. The mass exchanges between each sub-storage zone and the stream are parameterized by NEXSS in this model. Our approach is different from the approach in Painter (2018) which simulates one-dimensional advection-dispersion-reaction system for each storage zone. We agree with Dr. Painter that the different conceptualizations will have an impact on net biogeochemical processing and a discussion of these two different approaches is necessary.

---

## Author Comment (AC3) · 11 May 2020

The manuscript addresses the transport of nitrogen in river networks, and proposes an extension of the SWAT model that includes water exchange and biogeochemical reaction in the hyporheic zone. The model is applied to the watershed of the Columbia River. Proposing an extention of SWAT that includes the effect of hyporheic process is a valuable contribution to research in solute fate in catchments, as catchment-scale models usually do not encompass these processes. However, additional efforts are needed to clarify some methodological parts. The main issues are described below.

Response: We thank the reviewer for recognizing the value of our work and the constructive comments to improve our manuscript.

**Main comments:**

- ROLE OF GROUNDWATER: Vertical upwelling of groundwater through the hyporheic zone and to the stream seems not to be included (see detailed comments, line 205). The contribution of groundwater as lateral flow should be better explained (206). USE OF RESIDENCE TIME DISTRIBUTION: the authors explore the impact of using a distribution of residence time instead of an average value. This is a valuable attempt, but the way these distributions are chosen is unclear (251, 305).

Response: We apologize for the incomplete description of the boundary conditions and they will be added in the detailed comments below.

The details of how the residence times were chosen based on a given distribution is shown below:

we assume equal fraction for each sub-storage zone, i.e., a vertical or horizontal storage is evenly divided into  $N_s$  sub-storage zones.

To extract Ns residence times for the sub-storage zones in a given HZ with mean residence time  $\tau_m$ , we use the exponential distribution (P=1-e{-ts/tm). We have one distribution for the vertical HZ and one for the lateral HZ, with means equal to the vertical and lateral residence time calculated from NEXSS. Ns discrete residence time values with their corresponding probability were then extracted from this distribution for both HZs. For example, if Ns = 10, the fraction of each sub-storage zone is 0.1. The average residence time of each sub-storage corresponds to a value with probability less than or equal to 0.05, 0.15, 0.25, 0.35, 0.45, 0.55, 0.65, 0.75, 0.85, and 0.95, respectively. The minimum and maximum residence time are discrete values that are less than or equal to the given probability of 0.05 and 0.95, respectively. The maximum residence is interpolated from equation  $1-e^{(-ts/tm)}=0.95$ . The other discrete residence times are similarly solved.

**Detailed comments:**

**line 205 "the bottom boundary has a prescribed flux" -> how is it chosen? Is it equal to zero?**

Response: The bottom boundary is defined as  $q_u \approx 0.57 \text{K J}_y$  (Boano et al., 2009), where  $J_y$  is the mean head gradient across the alluvial valley.

Boano, F., Revelli, R. & Ridolfi, L. Quantifying the impact of groundwater discharge on the surface–subsurface exchange. Hydrol. Process. 23, 2108–2116 (2009)

206 "flow is solved by the vertically integrated groundwater flow equation with the Dupuit–Forchheimer assumption" -> for consistence, I suggest to briefly state boundary conditions also for determining lateral flow.

Response: The boundaries are stated below:

Following Gomez et al. (2012), the river is conceptualized as sinusoidal with wavelength  $\lambda$  [L] and amplitude  $\alpha$ [L]. A prescribed hydraulic head  $\psi_s(x) = \psi_0 + (J_x/\sigma)(s(x)$  is assigned along the river stretch, where  $\psi_0$  [L] is the elevation of the free surface elevation at the downstream end of the river above the horizontal bottom, s(x) [L] is the arc length along the boundary,  $\sigma = s(\lambda)/\lambda$  is sinuosity,  $J_x$  is the mean head gradient along the valley in the downstream direction. The boundary at a distance  $\lambda$  from the channel axis has a prescribed head  $\psi(x, y = \lambda) = J_x x + J_y \lambda$ , where Jy is the mean head gradient across the alluvial valley. The upgradient and downgradient boundaries of the domain along the reach are assumed periodic with a prescribed head drop  $\psi(x = 0, y) = \psi(x = 2\lambda, y) - 2\lambda J_x$ .

Gomez, J. D., J. L. Wilson, and M. B. Cardenas (2012), Residence time distributions in sinuosity-driven hyporheic 407 zones and their biogeochemical effects, Water Resources Research, 48, W09533, doi:10.1029/2012WR012180.

230 "Compared to other reaches in the watershed, the Columbia River is characterized by relatively larger exchange flow" -> Fig. 3 shows exchange flow values up to the order of 100 m3/s (vertical flow) and 1 m3/s (lateral flow). These values are extremely high and deserve some explanation on why they are considered realistic. It is possibly that they occur only on very long reaches, but I would check this and verify the values of flux per unit area that provide values whose magnitude can be more easily assessed.

Response: The relatively larger exchange flow is mainly due to the large size of a reach. We replotted Fig. 3 c,d as the values of flux per unit area for easy assessment as shown below.

247 "only one exchange rate [...] for each zone" -> I would state more clearly that two zones are used to represent vertical and lateral exchange, as it comes out later.

**Response: Yes, two zones are used to represent vertical and lateral exchange. Thanks for the suggestion.**

251 "1) replacing the residence time and exchange flux with those predicted by NEXSS using seasonal flow conditions; 2) replacing the single storage zone in vertical and lateral with sub-storage zones within a storage zone, assuming a distribution of residence time [...]" -> a few additional words would make easier to understand how these scenario have been built. Specifically, I recommend to specify 1) which "seasonal flow conditions" have been considered and 2) how the characteristics of the "distribution of residence time" have been determined. There is a mathematical explanation for it, but it is unclear how the specific values of tau\_s, j have been chosen. At present, some of (but not all) these information are provide in sections 4.3 and 4.4 of the Results' section, but they would better located here rather than among the Results. Alternatively, is should be at least anticipated that they are reported later.

Response: For the seasonal flow conditions, NEXSS model was run 12 times using mean monthly streamflow conditions. The exchange fluxes and residence times now change monthly instead of being constants throughout the whole simulation

Please see the detailed description of how specific residence time have been determined in the response to the main comments above.

288 "It's not true [the fact that lateral HZ are in dynamic steady state] for RCH77 and RCH88 as their exchange flows are much smaller" -> why only these reaches? RCH67, 53, 93, 100 and 101 all exhibit the same behavior.

Response: Thanks for catching this. After careful checking, a bug was found and fixed in the code. The bug mainly affects the magnitude in the lateral HZ, not the overall observations in the original manuscript. We corrected the figure and the statement:

They are also in dynamic steady state with the lateral HZ for reaches RCH27, RCH24, RCH28, and RCH20, RCH77 and RCH88 suggesting that lateral exchange can be important too. It's not true for RCH77 and RCH88 RCH53, RCH67, RCH93, RCH100, RCH101 as their residence times are much longer.

Line plots in Figures 4, 5C,D, 6, 8 are corrected. Please find the updates at the end of the response.

Fig. 5, caption: it should specified that this is the MRMT scenario. Same for other figures.

Response: Specified as:

Figure 5. Nitrate concentration in the stream and HZs along the Columbia River using MRMT.

**Figure 8. Nitrate concentration in the stream and vertical HZ at RCH27 (a),(b) and at the outlet (c),(d) with and without HZ perturbation using MRMT+BGC**

305 "Using exponential residence time distribution and 20 sub-storage zones, we had multiple rates based on the mean residence time from NEXSS. Assuming the exchange flux from the NEXSS estimation is equally distributed in each sub-storage zone, the residence time for each sub-sotrage zone is calculated using Eq. 12." -> this explanation is unclear. How has each residence time estimated with NEXSS (denoted here as T) transformed into multiple (20) residence times? From this description I imagine that for each zone an expontial distribution with mean equal to T was defined, and then 20 values with their corresponding probability were extracted from this distribution. Anyway, this is not fully clear from the text, and in any case many details are missing (e.g., how was the maximum residence time chosen?). I strongly recommend to provide more details about this part as it is fundamental to obtain a representative distribution of residence times. As a further notice, the choice of equally distributed fluxes is simple but debatable, as it is known from the classic theory of Elliott and Brooks (1997) that exchange flowpaths with higher fluxes penetrate deeper in the streambed and are hence characterized by longer times. I am unsure if this would entail a significant difference, but if it is feasible I recommend verifying the impact of this assumption.

**Response: Thanks for the suggestion and the reference. Please see the response to the main comment on how to choose residence times.**

We ran another simulation assuming higher exchange fluxes are associated with longer residence times and compared the impact of flux distribution based on different assumptions. Nitrate removal due to

denitrification in the HZs can be significantly reduced if larger flux associated with longer residence time is assumed compared to that with the assumption of equally distributed fluxes.

**307 "sotrage" -> "storage"**

**Response: Thanks for the correction!**

307 "Simulation with multiple exchange rates within each storage zone showed less removal of nitrate in the stream through microbial respiration in the HZs compared to the single-rate simulation (Fig. 7)." - > as far as I can see by eye, the difference is rather small. If so, I would mention it.

**Response: Agreed.**

Fig. 6: because all these scenarios include biogeochemical reactions, I recommend labeling them coherently, i.e., MRMT+BGC, SEASONAL MRMT+BGC.

**Response: Thanks for the suggestion.**

"high stream nitrate concentration than those shown in the bASE case can occur" -> is this a regular feature or has it only been observed occasionally?

Response: It seems to be a regular feature based on limited samples at a location close to the outlet.

**319 "bASE" -> "BASE"**

**Response: Thanks for the correction.**

325 "nitrate coming from these wasteways will exchange in the HZs in a short time and will not be expected to have a big impact on surface water quality" -> the link between residence time and impact on water quality is not so evident. What is clear from fig.10 is that in this reach HZ and river concentrations exhibit syncronous variations, as already expected from the previously shown results. I understand that if the residence time in the HZ is large enough then the increase in NO3 concentration due to the point source can be buffered and possibly attenuated, but the comments here do not clarify this well enough.

Response: Shorter residence time results in faster nitrate exchange rate (which is the inverse of residence time) between the stream and HZ. Faster exchange rate drives more stream nitrate into the HZ if stream nitrate concentration is high and increases the nitrate concentration in the HZ, which then increases the denitrification rate or nitrate consumption rate in the HZ. It can be better explained by the total nitrate consumption in the HZ.

341 "our simulations show that HZs can attenuate the peak nitrate concentrations in the stream" -> it would be useful to report a quantative assessment (e.g., concentration reduction between xxx and yyy %) instead of just sending back the reader to fig.4.

Response: There was a 11.6% of concentration reduction on average compared to the base case without MRMT.

375 "limations" -> "limitations"

Response: Thanks for the correction!

Figure 4.

---

## Author Response (AR1)

May 11, 2020

Dear Dr. Yool,

We are pleased to submit our revised manuscript entitled "A multirate mass transfer model to represent the interaction of multicomponent biogeochemical processes between surface water and hyporheic zones (SWAT-MRMT-R 1.0)" for your consideration in Geoscientific Model Development.

We thank the two anonymous reviewers for their positive feedbacks and constructive comments that helped us clarify the important aspects of our work.  We have addressed all the comments and revised the manuscript accordingly. Point-by-point responses are listed in the responses in blue.  The line numbers shown in the response are for the tracked version of the revision attached to this response.

Thanks for your consideration and we look forward to hearing from you.

Sincerely,

Yilin Fang and Co-authors

**Response to review comments**

**Anonymous Referee #1**

The discussion paper "A multirate mass transfer model to represent the interaction of multicomponent biogeochemical processes between surface water and hyporheic zones (SWAT-MRMT-R 1.0)" introduces a newly-developed model, named SWATMRMT-R, to simulate the hydrological and biogeochemical interaction between surface water and hyporheic zones. Hyporheic zone is usually ignored in large-scale watershed models because of its complex processes. Therefore, I think adding the simulation of hyporheic zone processes and connection hyporheic zones and surface water has a good intention and is appreciated. I have some major comments for this manuscript:

Response: We thank the reviewer for acknowledging the value of our model and the encouragements. We considered every comment when revising our manuscript. The following are our detailed responses to each comment:

1.The manuscript description about SWAT-MRMT-R, the connection between SWAT and MRMT, the connection of SWAT code and PFLOTRAN is not very clear. There is abundance of model concepts, software, and codes involving in this work (SWAT, MRMT, PFLOTRAN, NEXSS, NHD plus V2), and the connection between them was mentioned in different places in the manuscript. But I want to put in in a figure/schematization, to make the readers clear about the role of each of them in the model. I recommend modifying figure 1 or having a separate figure showing the connection between these.

Response: We only modified the in-stream nutrient transformation module in SWAT for the purpose of this study. We made clarifications in the revision (**lines 90-94**). The stream water column and a hyporheic zone at a reach in a river network can be conceptualized as separate batch reactors gaining or losing mass due to hydrologic exchange. NEXSS is only a tool used to estimate the hydrologic exchange fluxes and residence times for discretized reaches for a river network (NHD plus V2 in this study). The results from NEXSS were provided to us from our collaborator, and we used them as input to the model developed in this study. NEXSS is not required as long as there is a way or model to estimate the hydrologic exchange fluxes and residence times. The following schematics shows the connections between different reactors. The storage zones do not communicate with each other. The reactions within these batch reactors and mass transfer between the in-stream reactor and hyporheic (storage) reactors are solved using PFLOTRAN.

[Figure]

2. It is not very clear to me about the connection in hydrology between SWAT and MRMT. Does SWAT provide inflow of upland catchment to hyporheic zones which is represented by MRMT, and how the transmission losses through streambed calculated by SWAT affect the hyporheic zones? I think hydrological exchange between surface water and hyporheic zones is very important for the constituent exchange, so please make it clear.

Response: We agree that the hydrological exchange between surface water and hyporheic zones is very important for the constituent. MRMT is an extension of the commonly used transient storage model to represent riverine solute transport. It does not affect the hydrology part of SWAT. As SWAT solves solute and reactions using the operator splitting approach, we only modified the in-stream nutrient transformation module in SWAT for the purpose of this study, treating the hydrological exchange between the storage zone and stream as kinetic process or "kinetic reactions". We made clarifications (**lines 95-98**) in the revision.

 3. How many storage zones in your conceptualization of the hyporheic zones? Is the number of storage zones defined by the user in the setup? What is the maximum number of storage zones?

Response: The model is not limited by the number of hyporheic zones. The number of storage zones can be defined by the user in the setup. We added statement in the revision (**lines 97-98**).

4. Why can NEXSS not use the reach system generated by SWAT to calculate residence time and exchange flux?

Response: NEXSS can use different reach system as long as the values of bankfull channel width, discharge, median grain size, channel slope, sinuosity, and regional hydraulic head gradient along and across the reach are prescribed for each individual reach. We only used the NEXSS output from a previously generated model based on NHD river network through our collaborator and it is not our focus in this study. We made the clarifications in the revision (**lines 92-93**).

5. For matching the channel systems between NEXSS and SWAT, it seems that NEXSS reaches that do not overlap with those of SWAT are not considered. In the example in figure 2b, only reaches 386 and 380 contribute to SWAT reach 36. However, I think reach 374 also contributes to SWAT reach 36. If we change the threshold stream definition in SWAT, SWAT can generate a more detailed stream network and can overlap with all of the reaches from NEXSS.

Response: In the current configuration, NEXSS reaches that do not overlap with those of SWAT are not considered. We agree that a more detailed stream network can overlap with all of the reaches from NEXSS. We clarified it in the revision (**lines 237-239, 244-245**).

6. The paper shows several modelling experiments. But I wonder if the authors did compare the model predictions with field measurements. I did not see calibration/validation in this paper. I do think the comparison with measurements is very necessary for a modelling paper. So please add a session on this if data is available.

Response: We agree that the comparison with field measurements is necessary. We didn't compare with field measurements in this study as we don't have available data. But the comparison is planned when we have observations from field campaigns, which are ongoing. We pointed this out in the revision (**lines 437-438**).

7. Do you have a SWAT model application in this case study? Do you consider comparing SWAT-MRMT-R with SWAT to show the better prediction when hyporheic processes are included?

Response: We don't have model application yet. But it is planned in future study when field data are available. One of the advantages of the model is that it can be used to define the reaches that are biogeochemical hot spots before conducting the experiment and then combine the model with experiment data collected at those hot spots for a better mechanistic process understanding of river corridor functioning at the reach scale and model improvement (**lines 449-453**).

8. SWAT-MODFLOW coupling SWAT and MODFLOW models can represent hyporheic zones and its interaction with the stream. The author should add reference about SWAT-MODFLOW in the manuscript. What is the advantage of SWAT-MRMT-R compared to SWAT-MODFLOW? I suggest adding a section discussing about the strengths, weakness/limitations of the models, what cases the users should use SWAT-MRMT-R in the manuscript.

Response: We agree that a fully integrated model such as SWAT-MODFLOW can represent hyporheic zones, but it is computationally demanding when high resolution simulations are needed to capture the hyporheic zones that are highly variable in space and time. SWAT-MRMT-R we developed in this study is a simplified model that is easy to be adapted and is useful in providing insightful predictions of controls on time and locations of biogeochemical hot spots and river corridor functioning using physically derived parameters at the reach and watershed scales. We discussed the model strengths and limitations in the revision (**lines 37-41, lines 454-460**).

**Anonymous Referee #2**

The manuscript addresses the transport of nitrogen in river networks, and proposes an extension of the SWAT model that includes water exchange and biogeochemical reaction in the hyporheic zone. The model is applied to the watershed of the Columbia River. Proposing an extention of SWAT that includes the effect of hyporheic process is a valuable contribution to research in solute fate in catchments, as catchment-scale models usually do not encompass these processes. However, additional efforts are needed to clarify some methodological parts. The main issues are described below.

Response: We thank the reviewer for recognizing the value of our work and the constructive comments to improve our manuscript.

Main comments:

- ROLE OF GROUNDWATER: Vertical upwelling of groundwater through the hyporheic zone and to the stream seems not to be included (see detailed comments, line 205). The contribution of groundwater as lateral flow should be better explained (206). USE OF RESIDENCE TIME DISTRIBUTION: the authors explore the impact of using a distribution of residence time instead of an average value. This is a valuable attempt, but the way these distributions are chosen is unclear (251, 305).

Response: We apologize for the incomplete description of the boundary conditions and they will be addressed in the detailed comments below.  Details for the boundary conditions are added in the revision (**lines 218-222 and 223-230**).

The details of how the residence times were chosen based on a given distribution is shown below:

*we assume equal fraction for each sub-storage zone, i.e., a vertical or horizontal storage is evenly divided into $N_s$ sub-storage zones.*

*To extract $N_s$ residence times for the sub-storage zones in a given HZ with mean residence time $\tau_m$, we use the exponential distribution ($P=1-e^{(-\tau s/\tau m)}$}. We have one distribution for the vertical HZ and one for the lateral HZ, with means equal to the vertical and lateral residence time calculated from NEXSS. $N_s$ discrete residence time values with their corresponding probability were then extracted from this distribution for both HZs. For example, if $N_s = 10$, the fraction of each sub-storage zone is 0.1.  The average residence time of each sub-storage corresponds to a value with probability less than or equal to 0.05, 0.15, 0.25, 0.35, 0.45, 0.55, 0.65, 0.75, 0.85, and 0.95, respectively.  The minimum and maximum residence time are discrete values that are less than or equal to the given probability of 0.05 and 0.95, respectively. The maximum residence is interpolated from equation $1-e^{(-\tau s/\tau m)}= 0.95$. The other discrete residence times are similarly solved.*

We added more details on how the discretized residence times were chosen in the revision (**lines 303-313**).

Detailed comments:

line 205 "the bottom boundary has a prescribed flux" -> how is it chosen? Is it equal to zero?

Response: The bottom boundary is defined as $q_u \approx 0.57K\ J_y$ (Boano et al., 2009), where $J_y$ is the mean head gradient across the alluvial valley.  The definition is added in the revision (**lines 219-220**).

Boano, F., Revelli, R. & Ridolfi, L. Quantifying the impact of groundwater discharge on the surface–subsurface exchange. Hydrol. Process. 23, 2108–2116 (2009)

206 "flow is solved by the vertically integrated groundwater flow equation with the Dupuit–Forchheimer assumption" -> for consistence, I suggest to briefly state boundary conditions also for determining lateral flow.

Response: We added the following in the revision (**lines 223-230**):

*Following Gomez et al. (2012), the river is conceptualized as sinusoidal with wavelength λ [L] and amplitude α[L]. A prescribed hydraulic head $ψ_s(x) = ψ_0 + (J_x/σ)(s(x)$ is assigned along the river stretch, where $ψ_0$ [L] is the elevation of the free surface elevation at the downstream end of the river above the horizontal bottom, s(x) [L] is the arc length along the boundary, σ = s(λ)/λ is sinuosity, $J_x$ is the mean head gradient along the valley in the downstream direction. The boundary at a distance λ from the channel axis has a prescribed head $ψ(x,y = λ) = J_xx + J_yλ$, where Jy is the mean head gradient across the alluvial valley. The upgradient and downgradient boundaries of the domain along the reach are assumed periodic with a prescribed head drop $ψ(x = 0,y) =ψ(x = 2λ,y) − 2λJ_x$.*

*Gomez, J. D., J. L. Wilson, and M. B. Cardenas (2012), Residence time distributions in sinuosity-driven hyporheic 407 zones and their biogeochemical effects, Water Resources Research, 48, W09533, doi:10.1029/2012WR012180.*

230 "Compared to other reaches in the watershed, the Columbia River is characterized by relatively larger exchange flow" -> Fig. 3 shows exchange flow values up to the order of 100 m3/s (vertical flow) and 1 m3/s (lateral flow). These values are extremely high and deserve some explanation on why they are considered realistic. It is possibly that they occur only on very long reaches, but I would check this and verify the values of flux per unit area that provide values whose magnitude can be more easily assessed.

Response: The relatively larger exchange flow is mainly due to the large size of a reach. We replotted Fig. 3 c,d as the values of flux per unit area (shown below) for easy assessment.

[Figure]

247 "only one exchange rate [...] for each zone" -> I would state more clearly that two zones are used to represent vertical and lateral exchange, as it comes out later.

Response: Modified as suggested (**lines 273-274**).

251 "1) replacing the residence time and exchange flux with those predicted by NEXSS using seasonal flow conditions; 2) replacing the single storage zone in vertical and lateral with sub-storage zones within a storage zone, assuming a distribution of residence time [...]" -> a few additional words would make easier to understand how these scenario have been built. Specifically, I recommend to specify 1) which "seasonal flow conditions" have been considered and 2) how the characteristics of the "distribution of residence time" have been determined. There is a mathematical explanation for it, but it is unclear how

the specific values of tau_s,j have been chosen. At present, some of (but not all) these information are provide in sections 4.3 and 4.4 of the Results' section, but they would better located here rather than among the Results. Alternatively, is should be at least anticipated that they are reported later.

Response: For the seasonal flow conditions, NEXSS model was run 12 times using mean monthly streamflow conditions. The exchange fluxes and residence times now change monthly instead of being constants throughout the whole simulation.  We added the above in the revision (**lines 280-281).**

Please see the detailed description of how specific residence time have been determined in the response to the main comments above.  We added more details as suggested and moved up those described in the results' section to where it is introduced (**lines 303-313**).

288 "It's not true [the fact that lateral HZ are in dynamic steady state] for RCH77 and RCH88 as their exchange flows are much smaller" -> why only these reaches? RCH67, 53, 93, 100 and 101 all exhibit the same behavior.

Response: Thanks for catching this.  After careful checking, a bug was found and fixed in the code.  The bug mainly affects the magnitude in the lateral HZ, not the overall observations in the original manuscript. We corrected the figure and the statement as follows (**lines 325-327**):

*"They are also in dynamic steady state with the lateral HZ for reaches RCH27, RCH24, RCH28, and RCH20, RCH77 and RCH88 suggesting that lateral exchange can be important too. It's not true for RCH77 and RCH88 RCH53, RCH67, RCH93, RCH100, RCH101 as their residence times are much longer*."

Line plots in Figures 4, 5C,D, 6, 8 are also corrected.  Please find the updates at the end of the response.

Fig. 5, caption: it should specified that this is the MRMT scenario. Same for other figures.

Response: We specified MRMT as suggested in the caption for figures that apply.

305 "Using exponential residence time distribution and 20 sub-storage zones, we had multiple rates based on the mean residence time from NEXSS. Assuming the exchange flux from the NEXSS estimation is equally distributed in each sub-storage zone, the residence time for each sub-sotrage zone is calculated using Eq. 12." -> this explanation is unclear. How has each residence time estimated with NEXSS (denoted here as T) transformed into multiple (20) residence times? From this description I imagine that for each zone an expontial distribution with mean equal to T was defined, and then 20 values with their corresponding probability were extracted from this distribution. Anyway, this is not fully clear from the text, and in any case many details are missing (e.g., how was the maximum residence time chosen?). I strongly recommend to provide more details about this part as it is fundamental to obtain a representative distribution of residence times. As a further notice, the choice of equally distributed fluxes is simple but debatable, as it is known from the classic theory of Elliott and Brooks (1997) that exchange flowpaths with higher fluxes penetrate deeper in the streambed and are hence characterized by longer times. I am unsure if this would entail a significant difference, but if it is feasible I recommend verifying the impact of this assumption.

Response: We moved the description of how the residence times were chosen to section 3.5 in the revision and added more details (**lines 306-313**) and made changes accordingly in **lines 343-346**.

Thanks for the reference of Elliott and Brooks (1997). We ran another simulation assuming higher exchange fluxes are associated with longer residence times and compared the impact of flux distribution based on different assumptions on HZ nitrate removal (**lines 346-353, lines 423-426**). Nitrate removal

due to denitrification in the HZs can be significantly reduced if larger flux associated with longer time is assumed compared to that with the assumption of equally distributed fluxes. We replaced Fig. 7 with the spatial nitrate removal for better visualization and explanation.

[Figure]

Figure 7. Total nitrate consumption through HZ denitrification at the end of simulation using a single storage zone in both vertical and lateral HZs (a), change in nitrate consumption with 10 sub-storage zones in both vertical and lateral HZs compared to the simulation with single storage zone using uniform exchange flux (b) and nonuniform exchange flux (c).

307 "sotrage" -> "storage"

Response: Corrected. Thanks!

307 "Simulation with multiple exchange rates within each storage zone showed less removal of nitrate in the stream through microbial respiration in the HZs compared to the single-rate simulation (Fig. 7)." -> as far as I can see by eye, the difference is rather small. If so, I would mention it.

Response: We removed the line plot in Fig. 7, replaced it with spatial nitrate consumption difference plots between scenarios (plot shown above), and pointed out the small difference between the single rate and MRMT with the assumption of equally distributed fluxes.

Fig. 6: because all these scenarios include biogeochemical reactions, I recommend labeling them coherently, i.e., MRMT+BGC, SEASONAL MRMT+BGC.

Response: Modified as suggested.

318    "high stream nitrate concentration than those shown in the bASE case can occur" -> is this a regular feature or has it only been observed occasionally?

Response: It seems to be a regular feature based on limited samples at a location close to the outlet.

319    "bASE" -> "BASE"

Response: Corrected.

325 "nitrate coming from these wasteways will exchange in the HZs in a short time and will not be expected to have a big impact on surface water quality" -> the link between residence time and impact on water quality is not so evident. What is clear from fig.10 is that in this reach HZ and river concentrations exhibit syncronous variations, as already expected from the previously shown results. I understand that if the residence time in the HZ is large enough then the increase in NO3 concentration due to the point source can be buffered and possibly attenuated, but the comments here do not clarify this well enough.

Response: Shorter residence time results in faster nitrate exchange rate (which is the inverse of residence time) between the stream and HZ. Faster exchange rate drives more stream nitrate into the HZ if stream concentrations are high and increases the nitrate concentration in the HZ, which then increases the denitrification rate or nitrate consumption rate in the HZ. We replaced the line plot in Figure 10 with the difference in HZ nitrate consumption between the irrigation return flow case and the base case for a better explanation (**lines 375-387**). Denitrification in the HZs can remove more nitrate at RCH77 and RCH88 even though the nitrate loadings and HZ volumes are smaller compared to those at RCH100 because of the short residence time that promotes larger reaction extent in the HZs at those locations.

[Figure]

Figure 10. Increase of HZ nitrate consumption (kg) through denitrification due to irrigation return flow compared to case MRMT+BGC. Text in green shows the amount of nitrate consumption, pink solid circles are the locations of irrigation wasteways, and numbers in parentheses are reaches of interest.

341 "our simulations show that HZs can attenuate the peak nitrate concentrations in the stream" -> it would be useful to report a quantative assessment (e.g., concentration reduction between xxx and yyy %) instead of just sending back the reader to fig.4.

Response: We added the following statement (**lines 398-399**): "our simulations show that HZs can attenuate the peak nitrate concentrations in the stream with mass transfer and biogeochemical reactions, with 11.6% of concentration reduction on average compared to the base case without MRMT."

375 "limations" -> "limitations"

Response: Corrected. Thanks!

Figure corrections:

[Figure]

Figure 4.

[Figure]

Figure 5C,D

[Figure]

Figure 6

[Figure]

Figure 8

[revised manuscript text omitted]

---

## Referee Report (RR1)

**Manuscript Number:** gmd-2019-301

**Title:** A multirate mass transfer model to represent the interaction of multicomponent biogeochemical processes between surface water and hyporheic zones (SWAT-MRMT-R 1.0)

**Authors:** Yilin Fang, Xingyuan Chen, Jesus Gomez velez, Xuesong Zhang, Zhuoran Duan, Glenn E. Hammond, Amy E. Goldman, Vanessa A. Garayburu-Caruso, and Emily B. Graham

**Type:** Second Submission (First review by this referee)

**Recommendation:** Publish

**Overview:**

The manuscript describes the synthesis of three previously described models (NEXSS, SWAT and PFLOTRAN) into a coherent framework (SWAT-MRMT-R) that tracks hydrologic and biogeochemical processes at the watershed scale (and possibly larger) with reach scale resolution. Hyporheic processes, both transport and reactive, are explicitly included. The capabilities of the SWAT-MRMT-R model are demonstrated by its application, in a numerical study, to the Hanford watershed of the Columbia River. The fate of nitrogen solute is evaluated under several different scenarios.

**General comments:**

The study of solute fate at watershed and basin scale or even global scales has been an area of increasing interest of late and the Fang et. al. manuscript is a significant contribution to that effort. Overall, the manuscript is will written and well organized. Objectives are clearly stated, and the research has met the stated goals. Conclusions are well constrained by the observations and data. In fact, I would like to congratulate the authors for resisting the temptation (if such temptation was ever present) to describe this work as a predictive model. I see a bit of an alarming trend within our community to do a purely numerical study and call it a predictive model. This is, in my view, a significant overreach. Your work is a significant contribution as it stands. And with some development and field validation could evolve to a predictive model that has real-world impact.

My recommendation is to publish. I feel that the manuscript could be published as it stands, but I do have some suggestions and comments that are worth considering.

NEXSS is not explicitly integrated into your modeling framework. However, NEXSS or something quite similar is essential to the successful implementation of the model. NEXSS does not seem to be readily accessible (maybe I am looking in the wrong place). If your goal if to deploy this framework to other researchers, then it would be quite useful to either make NEXSS more readily available or to provide a bit of guidance as to how the necessary information could be synthesized for large scale studies.

Overall you do a good job of describing the limitations of your modeling framework. There is one other factor that I think deserves some mention, even though it may seem completely obvious. The way that this model is parameterized with multiple reaches each possibly with multiple storage zones, transfer rates and reaction rates, means that, at the outlet, there are possibly a multitude of parameterizations that can yield the "right" answer – most of them for objectively the wrong reason. As such, if this model is to be used in real-world, large-scale studies then validation at multiple points within the study area should be strongly considered and recommended.

I have some minor suggestions which are listed below.

**Specific comments (need to be addressed in the narrative or explain why there is no need to revise):**

1. Author listing – I believe that Velez should be capitalized.
2. Line 15: "A two-step reactions for denitrification and an aerobic respiration reaction are assumed to represent…" There is a mismatch in tense. Could be modified as: "A two-step reaction *sequence* for denitrification and an aerobic respiration reaction *is* assumed to represent…" or something similar.
3. Lines 92 – 95: the wording is a bit ambiguous.  I believe that you are saying:
   - solute reaction equations are solved by the Newton-Raphson method
   - hydrologic transport is solved by operator splitting
   Please clarify for readability.
4. Section 2.4: These are questions/suggestions about future efforts that likely arise from choices that were necessarily made to facilitate the present study.

   Does SWAT surface water module include denitrification include denitrification?  Recent studies have shown this to be significant in high-order, turbid streams.

   For future iterations of the model, it would be interesting to include $NO_3^-$ -→ $N_2O$↑ as this is a significant reaction in large rivers, mostly n the water column but also in the HZ.
5. Line 160 – 169: The transition from the discussion of SWAT to PFLOTRAN is a bit abrupt. Suggest the following revision:

   In SWAT, dissolved nutrients are transported with the water and those sorbed to sediments are allowed to be deposited with the sediments on the bed of the channel (Neitsch et al., 2011). PFLOTRAN is an open source, massively-parallel reactive multiphase flow and multicomponent transport code. It has well-established documentation (https://www.pflotran.org/documentation/). Nutrient transport and reactions in SWAT are solved sequentially. We modified the explicit time-stepping algorithm in the original code for in-stream chemistry so the resulting nonlinear system of equations from the transformations taking place within the stream water and storage zones are simulated simultaneously with the implicit time stepping through the Newton Raphson method in batch mode (i.e., no transport) of the PFLOTRAN (Lichtner et al., 2017) model.
6. Line 223: Replace "[L] is the elevation of the free surface elevation at" with "[L] is the free surface elevation at"
7. Line 277- 279:  For the seasonal flow scenario, the application of NEXSS is clearly defined.  For the BASE and MRMT is seems a bit ambiguous.  Is it annually averaged over the same period?  Maybe I missed the it earlier in the paper, but it would be useful to make a clear distinction at the same point in the manuscript.
8. For Figure 4 which compares the BASE scenario to the MRMT scenario, it would be useful to add markers to one of the traces so that it would be visually obvious that the two traces follow essentially the same path.

---

## Author Response (AR2)

July 10, 2020

Dear Dr. Yool,

Following the reviewers' comments and your recommendation, we have made technical corrections in our manuscript entitled "A multirate mass transfer model to represent the interaction of multicomponent biogeochemical processes between surface water and hyporheic zones (SWAT-MRMT-R 1.0)".

Attached please find our response to the reviewers' comments.

Sincerely,

Yilin Fang and Co-authors

**Response to review comments**

**Anonymous Referee #2**

My only further suggestion is related to the results of section 4.4, and specifically on the estimated differences on nitrate removal (+3% vs -40%) depending on the modeling assumptions of multiple exchange zones. Beside stating the results, it would be useful that the authors provide some comments on the relevance of these findings: for instance, if the flux-weighting is thought to be more correct, the results imply that other approaches tend to overestimate nitrate removal; if it is unclear which approach is more reliable, then the results show how uncertain it is to estimante nitrate turnover and demonstrate the need for more research on this aspect. It is not required to give a definite answer in the manuscript, but a comment should be provided.

Response: We thank the reviewer for the further suggestion! We added the following comment in section 4.4 of the revision:

"These results show that the assumption of the exchange flux associated with each sub-storage zone can have a significant effect on the estimation of HZ nitrate removal and there is a need for more research on the reliable estimation of these exchange fluxes."

There is also a misprint in the title of fig. 7a ("Nitriate").

Response: Corrected.

**Anonymous Referee #3**

**Manuscript Number: gmd-2019-301**

**Title:** A multirate mass transfer model to represent the interaction of multicomponent biogeochemical processes between surface water and hyporheic zones (SWAT-MRMT-R 1.0)

**Authors:** Yilin Fang, Xingyuan Chen, Jesus Gomez velez, Xuesong Zhang, Zhuoran Duan, Glenn E. Hammond, Amy E. Goldman, Vanessa A. Garayburu-Caruso, and Emily B. Graham

Type: Second Submission (First review by this referee)

**Recommendation: Publish**

**Overview:** The manuscript describes the synthesis of three previously described models (NEXSS, SWAT and PFLOTRAN) into a coherent framework (SWAT-MRMT-R) that tracks hydrologic and biogeochemical processes at the watershed scale (and possibly larger) with reach scale resolution. Hyporheic processes, both transport and reactive, are explicitly included. The capabilities of the SWAT-MRMT-R model are demonstrated by its application, in a numerical study, to the Hanford watershed of the Columbia River. The fate of nitrogen solute is evaluated under several different scenarios. General comments: The study of solute fate at watershed and basin scale or even global scales has been an area of increasing interest of late and the Fang et. al. manuscript is a significant contribution to that effort. Overall, the manuscript is will written and well organized. Objectives are clearly stated, and the research has met the stated goals. Conclusions are well constrained by the observations and data. In fact, I would like to congratulate the authors for resisting the temptation (if such temptation was ever present) to describe this work as a predictive model. I see a bit of an alarming trend within our community to do a purely numerical study and call it a predictive model. This is, in my view, a significant overreach. Your work is a significant contribution as it stands. And with some development and field validation could evolve to a predictive model that has real-world impact.

My recommendation is to publish. I feel that the manuscript could be published as it stands, but I do have some suggestions and comments that are worth considering.

**Response: We appreciate the reviewer's recognition of our work and helpful comments. We have included the comments in our revision.**

NEXSS is not explicitly integrated into your modeling framework. However, NEXSS or something quite similar is essential to the successful implementation of the model. NEXSS does not seem to be readily accessible (maybe I am looking in the wrong place). If your goal if to deploy this framework to other researchers, then it would be quite useful to either make NEXSS more readily available or to provide a bit of guidance as to how the necessary information could be synthesized for large scale studies.

**Response: NEXSS will be made available soon.**

Overall you do a good job of describing the limitations of your modeling framework. There is one other factor that I think deserves some mention, even though it may seem completely obvious. The way that this model is parameterized with multiple reaches each possibly with multiple storage zones, transfer rates and reaction rates, means that, at the outlet, there are possibly a multitude of parameterizations that can yield the "right" answer – most of them for objectively the wrong reason. As such, if this model is to be used in real-world, large-scale studies then validation at multiple points within the study area should be strongly considered and recommended.

**Response: We added the following in the conclusion:**

"It is worth noting that this model has a multitude of parameters at reach scale. For real-world, largescale studies model validation at multiple points within the study area is necessary." I have some minor suggestions which are listed below.

**Specific comments (need to be addressed in the narrative or explain why there is no need to revise):**

1. Author listing – I believe that Velez should be capitalized.

**Response: Corrected.**

2. Line 15: "A two-step reactions for denitrification and an aerobic respiration reaction are assumed to represent..." There is a mismatch in tense. Could be modified as: "A two-step reaction sequence for denitrification and an aerobic respiration reaction is assumed to represent..." or something similar.

**Response: Revised as suggested.**

3. Lines 92 – 95: the wording is a bit ambiguous. I believe that you are saying: - solute reaction equations are solved by the Newton-Raphson method - hydrologic transport is solved by operator splitting Please clarify for readability.

**Response: We clarified as follows:**

**"the operator-splitting approach in which the solute transport and the reaction steps are solved separately in SWAT"**

4. Section 2.4: These are questions/suggestions about future efforts that likely arise from choices that were necessarily made to facilitate the present study. Does SWAT surface water module include denitrification include denitrification? Recent studies have shown this to be significant in high-order, turbid streams. For future iterations of the model, it would be interesting to include NO3 -  $\rightarrow$  N2O $\uparrow$  as this is a significant reaction in large rivers, mostly in the water column but also in the HZ.

**Response: SWAT surface water module does not include denitrification. It will be considered in the future iterations of the model.**

5. Line 160 – 169: The transition from the discussion of SWAT to PFLOTRAN is a bit abrupt. Suggest the following revision:

In SWAT, dissolved nutrients are transported with the water and those sorbed to sediments are allowed to be deposited with the sediments on the bed of the channel (Neitsch et al., 2011). PFLOTRAN is an open source, massively-parallel reactive multiphase flow and multicomponent transport code. It has well-established documentation (https://www.pflotran.org/documentation/). Nutrient transport and reactions in SWAT are solved sequentially. We modified the explicit time-stepping algorithm in the original code for instream chemistry so the resulting nonlinear system of equations from the transformations taking place within the stream water and storage zones are simulated simultaneously with the implicit time stepping through the Newton Raphson method in batch mode (i.e., no transport) of the PFLOTRAN (Lichtner et al., 2017) model.

**Response: Revised as suggested.**

6. Line 223: Replace "[L] is the elevation of the free surface elevation at" with "[L] is the free surface elevation at"

Response: Replaced.

7. Line 277- 279: For the seasonal flow scenario, the application of NEXSS is clearly defined. For the BASE and MRMT is seems a bit ambiguous. Is it annually averaged over the same period? Maybe I missed the it earlier in the paper, but it would be useful to make a clear distinction at the same point in the manuscript.

Response: BASE case does not use NEXSS. We clarified the application of NEXSS for MRMT as "The mass transfer parameters were estimated by NEXSS using the long-term average flow conditions." at the same point.

8. For Figure 4 which compares the BASE scenario to the MRMT scenario, it would be useful to add markers to one of the traces so that it would be visually obvious that the two traces follow essentially the same path.

Response: BASE case line width in Fig4a is increased for a better visual effect.